# 3D-printed low-voltage-driven ciliary hydrogel microactuators

Zemin Liu[1,2], Che Wang[3], Ziyu Ren[1,4], Chunxiang Wang[1,2], Wenkang Wang[5,6], Jongkuk Ko[1,7], Shanyuan Song[5], Chong Hong[5], Xi Chen[8], Hongguang Wang[9], Wenqi Hu[5,8,10,11 ✉] & Metin Sitti[1,3 ✉]

Micrometre-sized, densely packed natural cilia that perform non-reciprocal 3D motions with dynamically tunable collective patterns are crucial for biological processes such as microscale locomotion[1], nutrient acquisition[2], cell trafficking[3–5] and embryonic and neurological development[6–8]. However, replicating these motions in artificial systems remains challenging given the limits of scalable, locally controllable soft-bodied actuation at the micrometre scale. Overcoming this challenge would enhance our understanding of ciliary dynamics, clarify their biological importance and enable new microscale devices and bioinspired technologies. Here we show a previously unrecognized fast electrical response of micrometre-scale hydrogels, induced by voltages down to 1.5 V without hydrolysis, with bending motions driven by ion migration across a nanometre-scale hydrogel network 3D-printed by two-photon polymerization, occurring within milliseconds. On the basis of these findings, we print gel microcilia arrays composed of a soft acrylic acid-co-acrylamide (AAc-co-AAm) hydrogel (modulus of approximately 1,000 Pa) that respond to electrical stimuli within milliseconds. Each microcilium measures 2–10 μm in diameter and 18–90 μm in height, achieving 3D rotational bending motion at up to 40 Hz, mirroring the geometry and dynamics of natural cilia. These gel microcilia maintain functionality after 330,000 continuous actuation cycles with less than 30% performance degradation. The gel microcilia arrays can be integrated on flexible polyimide substrates and fabricated at large scale using conventional lithography techniques. They also offer individual dynamic control by means of microelectrode arrays and enable fluid manipulation and particle transport at the micrometre scale.

In low-Reynolds-number fluidic environments, viscous forces dominate inertial forces, leading to the evolution of micrometre-scale cilia structures capable of dynamically regulating beating patterns for efficient and adaptive swimming, locomotion and environmental manipulation[9]. Individual cilia generate 3D non-reciprocal motions[10,11], whereas coordinated movements, such as metachronal waves[1,12], enable effective fluid transport and manipulation[13,14]. For example, starfish larvae[2], reef corals[15], *Paramecium*[16], ctenophores[17,18] and *Stentor coeruleus*[19,20] use coordinated cilia arrays for swimming, feeding predation and predator evasion. In mammals, ciliary flows support critical physiological processes[21–23], such as neural cell maturation[6,7], airway clearance[24,25], reproductive cell transport[3–5] and the establishment of embryonic asymmetry[8] (Supplementary Video 1). Replicating these dynamic features in microscale artificial systems holds the potential for quantifiable assessment of the importance of ciliary motion, advancing microactuation and microrobotics technologies and enabling biomedical innovations[21–23].

Artificial cilia have been realized across a broad range of sizes, numbers, motion degrees of freedom and dynamic performances, yet challenges persist in matching natural cilia. At the centimetre scale, sparse arrays of pneumatic cilia—typically comprising six actuators—generate 2D reprogrammable metachronal waves at approximately 0.25 Hz, but bulky cavity designs hinder further miniaturization[26]. At the millimetre and micrometre scales, tens of magnetic-field-actuated cilia achieve both 2D and 3D motions at frequencies up to 100 Hz; however, it is challenging to generate spatially heterogeneous, high-resolution magnetic fields to reconfigure the collective beating pattern[27–30]. At the micrometre scale, ultrasound-driven rigid cilia[31,32] and arrays of pH-sensitive cilia comprising hundreds of actuators[33] exhibit only basic mechanical actuation, with kinematics that falls short of biological

[1]Physical Intelligence Department, Max Planck Institute for Intelligent Systems, Stuttgart, Germany. [2]Department of Information Technology and Electrical Engineering, ETH Zürich, Zürich, Switzerland. [3]School of Medicine and College of Engineering, Koç University, Istanbul, Turkey. [4]School of Mechanical Engineering and Automation, Beihang University, Beijing, China. [5]Bioinspired Autonomous Miniature Robots Group, Max Planck Institute for Intelligent Systems, Stuttgart, Germany. [6]International Institute for Interdisciplinary and Frontiers, Beihang University, Beijing, China. [7]Department of Chemical and Biological Engineering, Gachon University, Seongnam, Republic of Korea. [8]Department of Mechanical and Aerospace Engineering, The Hong Kong University of Science and Technology, Kowloon, Hong Kong. [9]Max Planck Institute for Solid State Research, Stuttgart, Germany. [10]Division of Integrative Systems and Design, The Hong Kong University of Science and Technology, Kowloon, Hong Kong. [11]Cheng Kar-Shun Robotics Institute (CKSRI), The Hong Kong University of Science and Technology, Kowloon, Hong Kong. ✉e-mail: wenqi@ust.hk; msitti@ku.edu.tr

performance and without dynamic reprogramming. In densely packed micrometre-scale arrays, light-responsive liquid-crystal cilia enable individually addressable bending and twisting and survive 100 cycles without damage. However, the slow actuation (about 0.1 Hz) limits the fluid-pumping capability and their coordinated motion requires several light sources, complicating integration and control[34,35]. Electrostatically actuated microsystems incorporating hundreds of cilia enable 2D motions at around 200 Hz in non-conductive high-strength dielectric liquids but are incompatible with ionic solutions, thereby restricting their applicability in biologically relevant environments[36]. Electrochemically redox-driven thin-film cilia, fabricated at both millimetre and micrometre scales in hundreds, are limited to 2D motions at frequencies between 5 and 100 Hz and are only tested around 1,000 cycles, as repeated redox may degrade the actuators[37,38]. Although some cilia systems use soft substrates to improve compliance[28–30,37], most use rigid platforms, which are less conformable. Altogether, constraints in miniaturization, fast dynamics, motion degrees of freedom, scalable fabrication and actuator durability highlight the need for artificial cilia comparable with their biological counterparts.

## Hydrogel microactuator fabrication

In this study, we use two-photon polymerization (TPP) for 3D printing and tune its key processing parameters, such as slicing and hatching (Extended Data Fig. 1a (ii)), to reduce the pore size of the ionic hydrogel from tens of micrometres in conventional millimetre hydrogel actuators (Extended Data Fig. 1b (i)) to the nanometre scale (transmission electron microscopy (TEM) image shown in Fig. 1a (1)). This nanoscale porosity increases the effective surface area and expands the capacity of the electric double layer (EDL), thereby enhancing ion transport and electro-osmotic flow in ionic solutions[39–41] (bottom image in Fig. 1a (1)).

We use this direct miniature hydrogel actuator printing technique to fabricate AAc-co-AAm gel microcilia (2–10 μm in diameter and 18–90 μm in height) with the microelectrodes placed around them (Extended Data Figs. 2 and 3). Under 1.5-V potential, which is below the threshold for electrolysis reactions (≤1.5 V), these closely spaced 30–300-μm electrodes generate electric fields ranging from 5,000 V m⁻¹ to 50,000 V m⁻¹. Figure 1a (2) depicts the bending mechanism. In deionized (DI) water, dissociated $H^+$ ions from carboxylic acid groups migrate and accumulate in region 1, causing the hydrogel to shrink in this region and bend towards the cathode. By contrast, in physiological saline (0.15380 mol l⁻¹ NaCl), dominant $Na^+$ ions draw water molecules into region 1, swelling the hydrogel and bending it towards the anode. This ion-migration-induced dynamic motion can mimic the 3D rotation, reconfiguration and localized heterogeneous behaviour observed in mouse embryonic node cilia[8] (Extended Data Fig. 4, Supplementary Fig. 1 and Supplementary Videos 1–4). Figure 1a (3) illustrates the configuration of an AAc-co-AAm hydrogel microcilia array with the integrated microelectrodes around them. Details of the fast bending mechanism of the gel microcilia are discussed in the next section.

Figure 1b shows that the gel microcilia array can be fabricated on a flexible polyimide-based microelectrode substrate. Electrical signals within one actuation cell (four electrodes around it) can actuate the central gel cilium to generate reprogrammable 3D bending motions (Supplementary Video 5). When in physiological saline, activating the left and right electrodes can bend the cilium in the *y*-direction (Fig. 1c (1)) and activating the top and bottom electrodes causes bending in the *x*-direction (Fig. 1c (3)). Furthermore, rhythmic electrode cycling can generate rotating electric fields, enabling anticlockwise (Fig. 1c (2)) or clockwise rotation (Fig. 1c (4)). Finally, many such gel microcilia, each with independently controlled electrodes, can form a programmable hydrogel cilia surface. Such an approach can be demonstrated by an array of 25 individually controlled gel microcilia (Fig. 1d (1)), an array of 625 gel microcilia (Fig. 1d (2)) or an array of 10⁶ gel microcilia (Fig. 1d (3)).

## Fast bending mechanism

Previously reported millimetre-scale hydrogels are actuated through interfacial pH or osmotic gradients[42–44]. By contrast, the micrometre-scale hydrogels reported here are actuated through internal ion migration by means of nanometre-scale pores, resulting in distinct bending behaviour and a 100-fold increase in bending speed response (Extended Data Figs. 5 and 6 and Supplementary Videos 6 and 7). This section will first explain the bending mechanism, particularly bending direction and bending amplitude, and then the fundamental reason for fast bending dynamics of the proposed hydrogel.

In this study, the gel microcilia exhibit different bending directions in solutions with different NaCl concentrations. In $H^+$-dominated DI water, they bend towards the cathode (Fig. 2a (1) and Supplementary Videos 8 and 9). In $Na^+$-dominated physiological saline (0.15380 mol l⁻¹ NaCl), they bend oppositely towards the anode (Fig. 2b (2) and Supplementary Video 8). At an intermediate concentration (0.00769 mol l⁻¹ NaCl), competitive $H^+/Na^+$ effects cause transient cathode bending followed by anode reversal (Fig. 2b (1) and Supplementary Video 8).

The bending direction can be explained by internal ion migration. In the $H^+$-dominated DI water, the $H^+$ ions from carboxylic acid groups migrate under an electric field and accumulate in the hydrogel network region 1 (ref. 45) (Fig. 1a (2)), causing local shrinkage and bending the gel microcilia towards the cathode. The AAc-co-AAm hydrogel is a pH-sensitive hydrogel that will shrink in acidic environments. The elevated $H^+$ concentration in low-pH conditions converts some fixed –COO⁻ groups to –COOH, reducing repulsive forces in the hydrogel network and causing the hydrogel to shrink[42,46]. Consequently, the gel microcilia bend towards the cathode in DI water owing to $H^+$ ion migration inside the hydrogel, whereas the previous millimetre hydrogel bends towards the anode owing to interfacial pH gradients[42–44] (Fig. 2a (1) and Extended Data Fig. 5).

Notably, increasing the –COOH concentration in the gel microcilia does not lead to larger bending amplitudes. We investigated this counter-intuitive behaviour by performing step response tests on gel microcilia with 15, 30, 45 and 60 wt% AAc (Fig. 2a (2)–(5)). The tests were conducted in DI water and detailed hydrogel compositions are provided in Methods. Under an applied electric field, the cilia tip bent towards the cathode (Fig. 2a (2)–(5) and Supplementary Video 8). Increased AAc concentration introduces more –COOH groups to the hydrogel network. However, higher AAc concentrations reduce bending angles (Fig. 2a (2)–(5)).

These experimental findings are supported by the fully coupled electro-chemo-mechanical simulations of AAc-based hydrogels. The model incorporates the influence of densely distributed –COO⁻ groups on ion migration within the hydrogel, capturing the influence of the nanometre-scale hydrogel pores. Full details of the modelling are provided in Methods. Consistent with the experiments, the simulated bending amplitude decreases from 21.5 μm to 15.1 μm as the AAc concentration increases from 15 wt% to 60 wt% (Fig. 2a (6)–(9)). This trend indicates that, in the electro-chemo-mechanical coupled system, the deleterious effects of higher AAc concentration (for example, increased Young's modulus (Extended Data Fig. 7a)) outweigh its beneficial contributions (for example, enhanced charge density within the hydrogel network).

When the hydrogel operates in $Na^+$-dominated physiological saline (0.15380 mol l⁻¹ NaCl), $Na^+$ ions can swell the hydrogel network region 1 (Fig. 1a (2)) and bend the gel microcilia towards the anode, in contrast to the $H^+$ ion effects (Fig. 2a). The fixed negatively charged –COO⁻ groups in the hydrogel network lead to a lower concentration of mobile Cl⁻ ions within the hydrogel compared with $Na^+$ ions[47]. When an electric field is applied, $Na^+$ ions migrate towards the cathode and accumulate in region 1 (refs. 43,45) (Fig. 1a (2)). This migration inside the hydrogel drags water molecules along with the $Na^+$ ions, causing region 1 (Fig. 1a (2)) to swell[48], which in turn bends the microcilia towards the anode

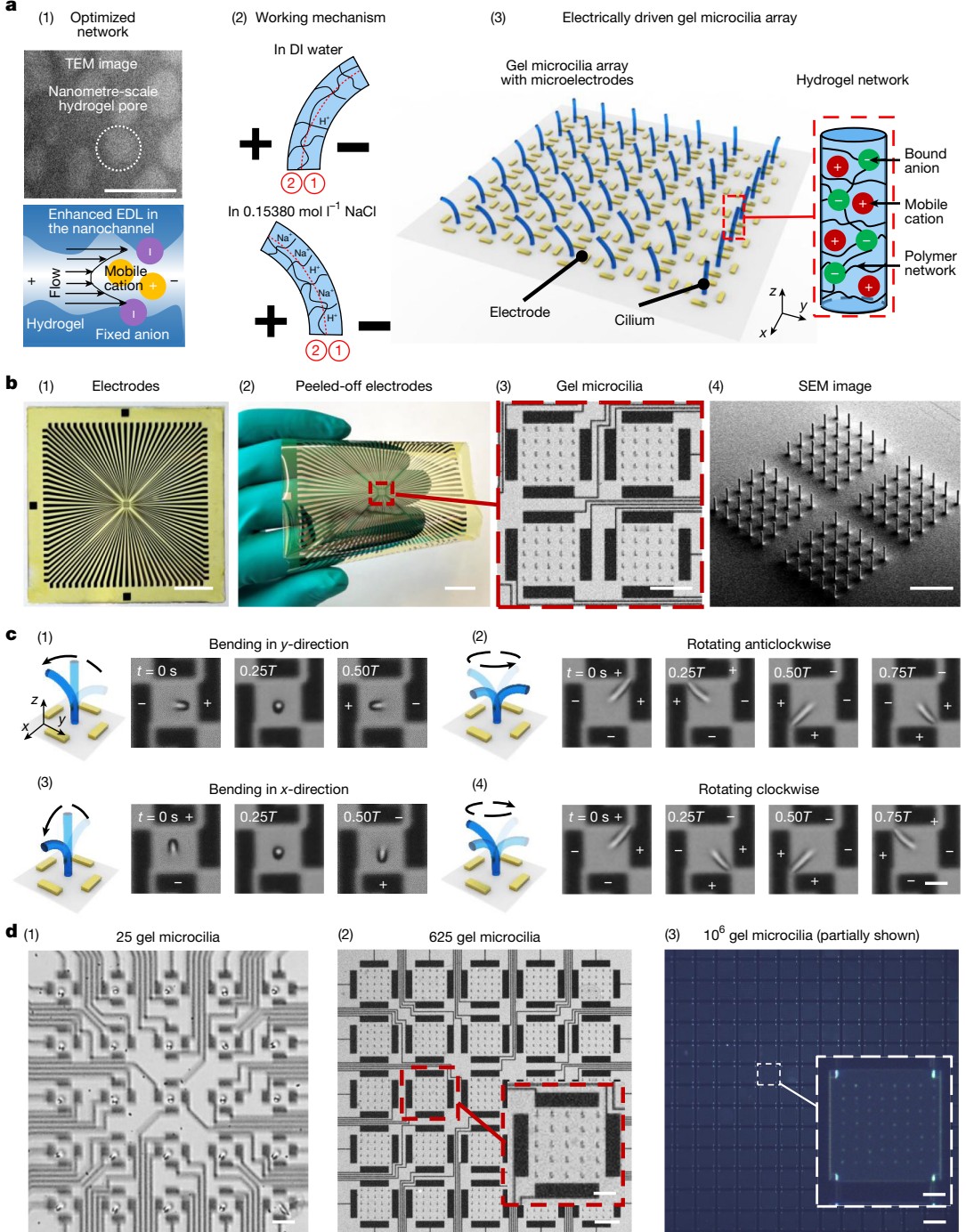

**Fig. 1 | Electrically driven hydrogel microactuator fabrication, mechanism and devices. a**, (1) Optimization of hydrogel pore size at the nanometre scale using TPP-based 3D printing. Top, TEM image of the 3D-printed hydrogel. Bottom, schematic showing enhanced ion flux and flow from increased EDL overlap in a nanometre-scale hydrogel channel under an electric field. (2) Working mechanism of the hydrogel microactuator. The hydrogel network is divided into region 1 (near the cathode) and region 2 (near the anode). In DI water, dissociated $H^+$ ions from the –COOH groups dominate. Under an electric field, concentrated $H^+$ in region 1 convert fixed –COO⁻ groups to –COOH, reducing repulsion and shrinking the network, bending the microactuator towards the cathode. In physiological saline, $Na^+$ ions dominate; concentrated $Na^+$ in region 1 attracts water, swelling the network and bending towards the anode. (3) Schematic of the electrically driven gel microcilia array and AAc-co-AAm hydrogel network structure. **b**, Gel microcilia array on a polyimide-based microelectrode substrate. (1) Microelectrodes on glass. (2) Flexible substrate

with microelectrodes on a human hand. (3) Gel microcilia array composed of four actuation cells, with each cell surrounded by four electrodes. (4) SEM image of the gel microcilia. **c**, Kinematics of a gel cilium in physiological saline (diameter 2 μm, height 18 μm; 5 Hz). The electrode polarity is marked by + and −. (1) Unidirectional bending along the $y$-direction. (2) Non-reciprocal 3D anticlockwise rotation. (3) Unidirectional bending along the $x$-direction. (4) Non-reciprocal 3D clockwise rotation. $T$ denotes one motion cycle. **d**, Main devices used in this work. (1) 25 gel microcilia (cilium diameter 10 μm, height 90 μm). Each cilium has four surrounding electrodes for individual control. (2) 625 gel microcilia (cilium diameter 10 μm, height 90 μm). Each actuation cell with 25 cilia exhibits synchronized motion (inset, one cell). (3) $10^6$ gel microcilia fabricated by micromoulding, partially shown in this image (cilium diameter 5 μm, height 35 μm; inset, one cell). Scale bars, 100 nm (**a** (1)); 2 cm (**b** (1), (2)); 200 μm (**b** (3), (4)); 6 μm (**c** (4)); 40 μm (**d** (1)); 200 μm (**d** (2)); 100 μm (**d** (2) inset); 300 μm (**d** (3)); 60 μm (**d** (3) inset).

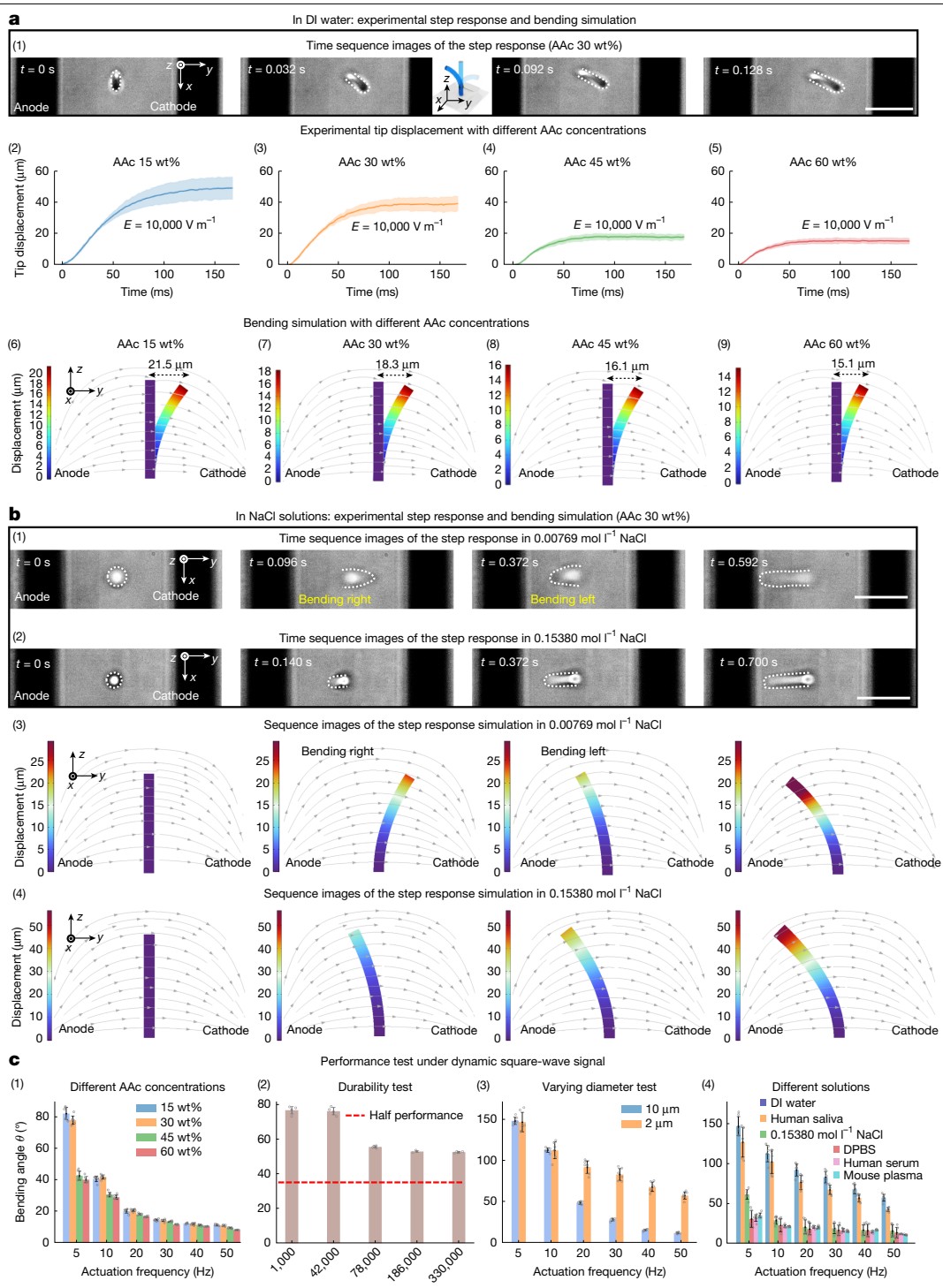

**Fig. 2 | Characterization of the gel microcilia actuator dynamics. a**, Step response of gel microcilia in experiments and corresponding bending simulations in DI water. (1) Time sequence of 30 wt% AAc gel microcilia bending in DI water with the left electrode as the anode and the right electrode as the cathode; dashed lines highlight cilia outlines. Under the field, cilia bend towards the cathode owing to $H^+$ accumulation on the right side (region 1 in Fig. 1a (2)). (2)–(5) Experimental tip displacement at different AAc concentrations shows reduced bending with higher AAc content (mean ± s.d.; $n = 6$ samples). (6)–(9) Simulated bending for varying AAc content reproduces the same trend, with displacement decreasing from 21.5 µm to 15.1 µm. **b**, Step response and simulations in NaCl solutions. (1) In 0.00769 mol $l^{-1}$ NaCl, cilia first bend towards the cathode as fast $H^+$ ions shrink the right side and then reverse towards the anode as slower $Na^+$ ion swelling dominates on the

right side. (2) In 0.15380 mol $l^{-1}$ NaCl, bending occurs only towards the anode. (3), (4) Simulations reproduce these bending behaviours in NaCl solutions. **c**, Influence of different factors on dynamic performance under a square-wave signal. (1) AAc concentration: lower AAc content enhances bending (mean ± s.d.; $n = 6$ samples). (2) Actuation cycle: the gel microcilia maintained a bending angle of 50° after 330,000 continuous actuation cycles, corresponding to 70% of the initial performance, and then stabilize (mean ± s.d.; $n = 4$ samples). (3) Cilium diameter: 2-µm actuators outperform 10-µm actuators at high frequencies (mean ± s.d.; $n = 6$ samples). (4) Solution type: gel microcilia tested in DI water, physiological saline, DPBS, human saliva, serum and mouse plasma (mean ± s.d.; $n = 6$ samples for DI and $n = 5$ samples for the others). Scale bars, 100 µm (**a** (1)); 100 µm (**b** (1), (2)). All of the characterization experiment details are provided in Supplementary Note 7.

(Fig. 2b). However, previously reported millimetre hydrogel bends towards the cathode under the same aqueous conditions owing to gel–solution interfacial osmotic pressure[42–44] (Extended Data Fig. 5).

To show this competitive $H^+$-driven shrinking and $Na^+$-driven swelling process, we examine the step response of 30 wt% AAc hydrogels in 0.00769 mol $l^{-1}$ and 0.15380 mol $l^{-1}$ NaCl solutions. The bending in the 0.00769 mol $l^{-1}$ NaCl solution (Fig. 2b (1)) contrasts with the bending in DI water (Fig. 2a (1)); the gel cilium initially bends towards the cathode and then gradually shifts to bend towards the anode (Fig. 2b (1) and Supplementary Video 8). This can be explained by the higher mobility of $H^+$ ions ($\mu_H = 3.62 \times 10^{-7}$ $m^2$ $s^{-1}$ $V^{-1}$) than that of $Na^+$ ions ($\mu_{Na} = 5.19 \times 10^{-8}$ $m^2$ $s^{-1}$ $V^{-1}$)—act first to accumulate in region 1 (Fig. 1a (2)), causing initial shrinkage and bending towards the cathode. $Na^+$ ions and water molecules subsequently enter the region, overcoming the initial shrinkage and bending towards the anode (Fig. 2b (1) and Supplementary Video 8).

At 0.15380 mol $l^{-1}$ NaCl, the gel cilium bends only towards the anode (Fig. 2b (2) and Supplementary Video 8). This behaviour is because, at higher NaCl concentrations, the swelling effect of $Na^+$ ions on the hydrogel network dominates the motion. Our simulations validate this mechanism: in 0.00769 mol $l^{-1}$ NaCl solution, the hydrogel initially bends towards the cathode and subsequently reverses direction towards the anode, consistent with the experimental observations (Fig. 2b (3)). By contrast, in 0.15380 mol $l^{-1}$ NaCl, both simulations and experiments show bending only towards the anode (Fig. 2b (2) and (4) and Supplementary Video 10). Together, these results confirm the roles of $H^+$ ions and $Na^+$ ions in driving the motion of microscale hydrogel actuators. Such bending happens very fast, with a step response time of 0.1–0.7 s.

Fast bending dynamics arise owing to two reasons. The first is that the expanded effective surface area of the EDL enhances ion transport. Particularly, TPP printing with carefully engineered polymerization pathways (Extended Data Fig. 3) and low printing power (about 15 mW) achieves nanometre-scale porosity while maintaining well-defined morphological features. The second is the fast $H^+$ and $Na^+$ migration at the micrometre scale. For instance, under a 10,000 V $m^{-1}$ electric field, achieved with 1.5 V applied over 150 μm, the $H^+$ ions (ion mobility $\mu_H = 3.62 \times 10^{-7}$ $m^2$ $s^{-1}$ $V^{-1}$) traverse a 10-μm distance in 2.8 ms, whereas it takes 16.7 ms for $Na^+$ ions ($\mu_{Na} = 5.19 \times 10^{-8}$ $m^2$ $s^{-1}$ $V^{-1}$). For a 2-μm distance, this decreases to 0.6 ms for $H^+$ and 3.4 ms for $Na^+$. The 10-μm and 2-μm distances correspond to the gel microcilia diameters in this work.

Because of these two reasons, these millisecond-scale ion migrations can trigger fast ion redistribution and subsequent mechanical responses, enabling rotational bending motions at frequencies up to 40 Hz. Extended Data Table 1 summarizes the performance of this work in comparison with previously reported ciliary actuators[26–38].

## Cilia bending characterization

Bending dynamics are critical for ciliary actuation performance characterization[26,28,38]. We first evaluate the impact of AAc concentration (15, 30, 45, 60 wt%) on gel microcilia bending dynamics in DI water (Fig. 2c (1)). Gel microcilia with lower AAc content (15 wt%) exhibit superior bending amplitudes at low frequencies (for example, twice the amplitude of 60 wt% AAc at 5 Hz), but this advantage diminishes at higher frequencies (for example, only 20% larger than the 60 wt% AAc at 50 Hz (Fig. 2c (1))). The dimensionless sperm number (Sp) is widely used to describe the reduction in bending at higher frequencies in low-Reynolds-number conditions (Re ≈ 0.001 in our system). As frequency increases, Sp also increases (Extended Data Fig. 7b), indicating that viscous effects become more dominant, leading to faster decay of bending along the actuator length and reduced tip displacement.

Durability testing under dynamic signals reveals sustained functionality: a single cilium still performs 50° bending after 330,000 cycles of continuous actuation (20 Hz for 5 h; Supplementary Video 11), which is 70% of the original performance (Fig. 2c (2)). The gel microcilia actuator remains functional after such extensive actuation, highlighting their durability. This longevity is attributed to the actuation mechanism, which relies on ion migration without chemical reactions. The overall device lifespan is primarily limited by the stability of the thin-film microelectrodes rather than the hydrogel actuator itself, as prolonged actuation often leads to electrode delamination from the substrate (see Supplementary Note 12). By contrast, previously proposed electrically driven cilia depend on repeated electrochemical oxidation and reduction, causing degradation and limiting their lifespan to several thousand cycles[37,38].

Gel microcilium dimensions, especially the diameter, also affect the bending performance. Gel microcilia with smaller diameter demonstrate better bending performance at high frequencies. As shown in Fig. 2c (3), under low-frequency signals, 2-μm-diameter and 10-μm-diameter hydrogel cilia show similar bending amplitudes. Both cilia approach a bending angle of approximately 150°, nearly the maximum achievable motion, indicating that each has ample time per actuation cycle to fully deform. Consequently, the benefits of smaller cilium do not manifest at low frequencies and the 2-μm-diameter cilium shows no distinct advantage in this regime. As the frequency increases, the 2-μm-diameter cilium gradually outperforms its 10-μm counterpart. At 50 Hz, its bending amplitude is nearly five times larger than that of the 10-μm cilium, as smaller hydrogel diameter needs shorter time for the ions to redistribute under a dynamic signal (see Supplementary Note 11).

As well as diameter, higher electric-field strength also accelerates ion transport and thereby enhances the performance of the actuator (Extended Data Fig. 7c). At a fixed diameter, taller hydrogel actuators exhibit improved performance owing to their increased effective actuation length (Extended Data Fig. 7d). However, fabricating high-aspect-ratio soft actuators can lead to structural collapse. Furthermore, increasing either the photoinitiator concentration or the printing power reduces the performance of the hydrogel actuators (Extended Data Fig. 7e,f) owing to the increased stiffness.

Finally, we test the performance of gel microcilia in three aqueous solutions (DI water, physiological saline (0.15380 mol $l^{-1}$ NaCl) and Dulbecco's phosphate-buffered saline (DPBS)) and three physiological fluids (human saliva, human serum and mouse plasma). As shown in Fig. 2c (4), actuation performance decreases across these environments, with high performance in DI water and human saliva, moderate performance in physiological saline and low performance in DPBS, human serum and mouse plasma. This trend aligns with the increasing ionic complexity of the solutions. DI water and human saliva feature a $H^+$-dominated environment, whereas physiological saline introduces $Na^+$ and $Cl^-$ ions. DPBS and the other two physiological fluids further complicate the system with balanced concentrations of more ions (for example, $Mg^{2+}$, $K^+$), creating competitive interactions between the ions of varying size, charge, mobility and hydration capacity. For instance, highly mobile ions rapidly migrate to region 1, inducing localized swelling, whereas less mobile ions are still in region 2 (Fig. 1a (2)), promoting counteractive swelling. This spatial and temporal mismatch in ion-driven swelling reduces overall bending performance. Nevertheless, hydrogel cilia remain functional in simple and complex ionic environments, including physiological fluids from human and mouse, showing their potential use in physiological fluid mixing (Supplementary Video 8).

## Reprogrammable motions

We demonstrate versatile control of coordinated motion across four gel microcilia systems under electrical stimulation. 2 gel microcilia (diameter 2 μm, height 18 μm) are programmed to perform synchronized unidirectional bending (Fig. 3a (1)) and 180° out-of-phase unidirectional bending (Fig. 3a (2)). By controlling electric signals across patterned electrodes, this cilia device can achieve synchronized clockwise 3D

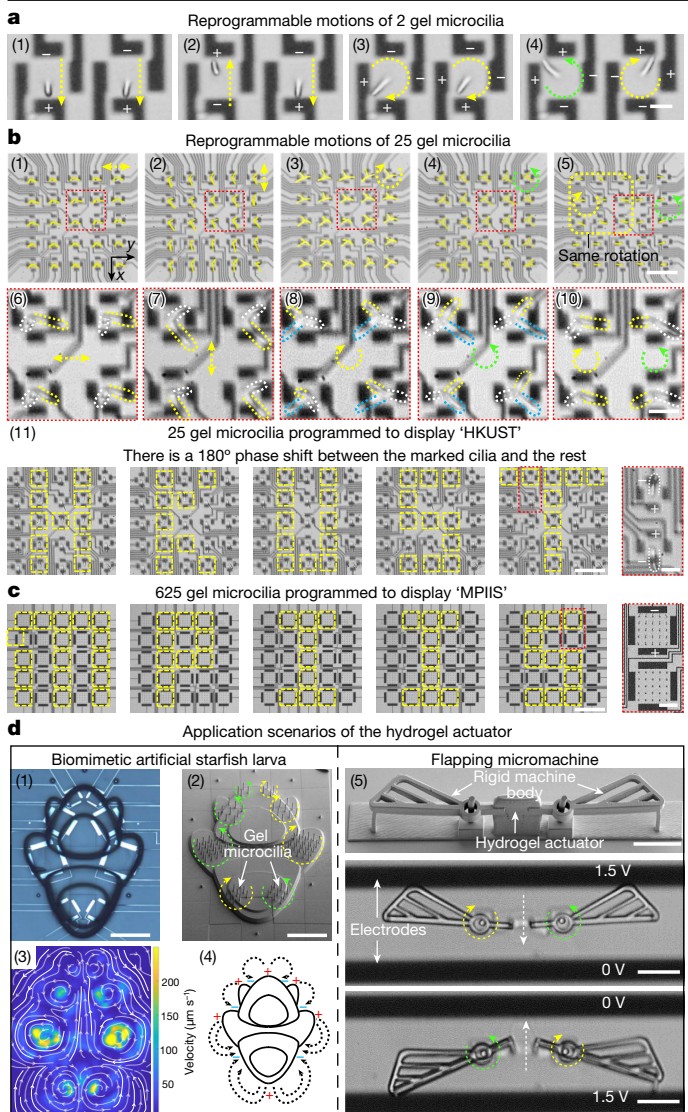

**Fig. 3 | Dynamic bending motions of gel microcilia arrays. a**, Reprogrammable motions of 2 gel microcilia (diameter 2 µm, height 18 µm; physiological saline; 5 Hz). (1) Synchronized unidirectional bending. (2) Unidirectional bending with 180° phase shift. (3) Synchronized clockwise 3D rotation. (4) Counter-rotating motion, left anticlockwise and right clockwise. **b**, Reprogrammable motions of an array of 5 × 5 gel microcilia (diameter 10 µm, height 90 µm; DI water). Yellow-shaded *z*-stack images show cilia motions. (1) Synchronized bending along the *y*-direction. (2) Synchronized bending along the *x*-direction. (3) Clockwise 3D rotation. (4) Anticlockwise 3D rotation. (5) 3 × 3 subarray (yellow dashed box) rotates clockwise, others anticlockwise. (6)–(10) Zoomed-in *z*-stack views corresponding to (1)–(5); cilia at identical time points share the same colour. (11) 'HKUST' displayed by an independently controlled 5 × 5 array; zoom-in shows opposite bending of adjacent cells in the 'T'. Motion frequencies, 10 Hz ((1)–(4)); 20 Hz (5); 5 Hz (11). **c**, 'MPIIS' displayed by a 25 × 25 array (diameter 10 µm, height 90 µm; DI water; 5 Hz). Zoom-in shows the upper cilium bending upward, whereas the lower remains stationary. **d**, Applications of hydrogel actuators. Left, biomimetic artificial starfish larva. Right, flapping micromachine. (1) Photo of patterned electrodes before cilia integration. (2) SEM image with arrows indicating cilia rotation. (3) PIV flow field generated by cilia motion. (4) Schematic showing the flow generated by biological starfish larva. (5) Flapping mechanism with integrated hydrogel actuators. The first row is the SEM image, followed by video frames showing the flapping motion. The electrode polarity is indicated by + and − in **a**, **b** (11) and **c**. Because **a** uses physiological saline and **b**–**d** use DI water, the bending direction differs. Scale bars, 8 µm (**a** (4)); 150 µm (**b** (5)); 40 µm (**b** (10)); 150 µm (**b** (11)); 30 µm (**b** (11) zoom-in); 750 µm (**c**); 100 µm (**c** zoom-in); 400 µm (**d** (1)); 360 µm (**d** (2)); 100 µm (**d** (5)).

rotational bending under uniform input signals (Fig. 3a (3)) or opposing rotation directions between gel microcilia through applying different input signals to the actuation cells (Fig. 3a (4) and Supplementary Video 12).

The coordinated motions of the gel microcilia can be extended to 5 × 5 and 25 × 25 arrays. An array of 5 × 5 gel microcilia (diameter 10 µm, height 90 µm) demonstrates individual addressability, enabling synchronized unidirectional bending, clockwise/anticlockwise rotation and independent actuation (Fig. 3b (1)–(10) and Supplementary Video 13). The array can also be reprogrammed to display patterns such as 'HKUST' (Fig. 3b (11) and Supplementary Video 14), with a 180° phase difference between the letters and the background gel microcilia enhancing visual contrast. Then, an array of 25 × 25 gel microcilia (diameter 10 µm, height 90 µm) is programmed to display the letters 'MPIIS' (Fig. 3c and Supplementary Video 15). Each actuation cell synchronizes 25 cilia, validating control over dense configurations.

Our hydrogel precursor and microactuation system are also compatible with standard fabrication methods, as demonstrated by the fabrication of $10^6$ gel microcilia using conventional lithography (Supplementary Figs. 2–6). Each actuation cell controls approximately 64 cilia, enabling synchronized bending along the *x*-axis or the *y*-axis (Supplementary Fig. 7 and Supplementary Video 16).

As shown in Extended Data Fig. 8a,b, hydrogel actuators, together with polyimide substrates and microelectrodes, were patterned onto a 3D hill-shaped surface with a height difference of 100 µm (see the scanning electron microscopy (SEM) images in Extended Data Fig. 8a,b), which closely resembles the non-flat morphology of physiological tissue surfaces. Controlled actuation enabled directed fluid transport on such 3D surfaces, including flows from right to left (Extended Data Fig. 8a and Supplementary Video 17) and from top to bottom (Extended Data Fig. 8b and Supplementary Video 18). Also, the actuators can be configured into alternative geometries. For example, integration with a 3D pyramid frame structure (Extended Data Fig. 8c and Supplementary Video 19) and direct printing into a helical shape (Extended Data Fig. 8d and Supplementary Video 20) demonstrated controlled ciliary flow manipulation under electrical stimulation.

To explore the application scenarios of this hydrogel microactuator, we fabricate an artificial starfish larva (Fig. 3d (1)–(4) and Supplementary Video 21), in which a biological starfish larva generates complex vortex arrays through ciliary motion to facilitate feeding and other essential processes[2]. The active boundary conditions arising from cilia–environment interactions may exert important effects in both biological and synthetic systems[2]. However, existing studies have been largely qualitative, based on biological observations without systematic quantification[2,20]. To address this gap, we fabricate an artificial starfish larva platform by combining microscale 3D printing of the larval body, microelectrode patterning on its curved surface and integration of hydrogel cilia. As shown in Fig. 3d (1)–(4) (Supplementary Video 21), the artificial starfish larva reproduces biologically analogous vortex arrays under electrical control, providing a reliable robotic platform for quantitative investigation of biomimetic processes and active boundary conditions[2].

Finally, we demonstrate the integration of hydrogel microactuators with micromechanical structures (Extended Data Fig. 8e and Fig. 3d (5)). Here electrically induced hydrogel bending is transduced into rotary and flapping motions through mechanical linkages, thereby broadening the scope of potential applications (Supplementary Videos 22 and 23). This capability highlights the potential of hydrogel microactuators to interface with complex micromachine architectures, expand microscale actuation strategies and advance the fields of microrobotics and microdevices.

## Dynamic fluid manipulation

In nature, biological cilia perform diverse functions by manipulating surrounding fluids, including airway clearance[25], gamete transport[3] and

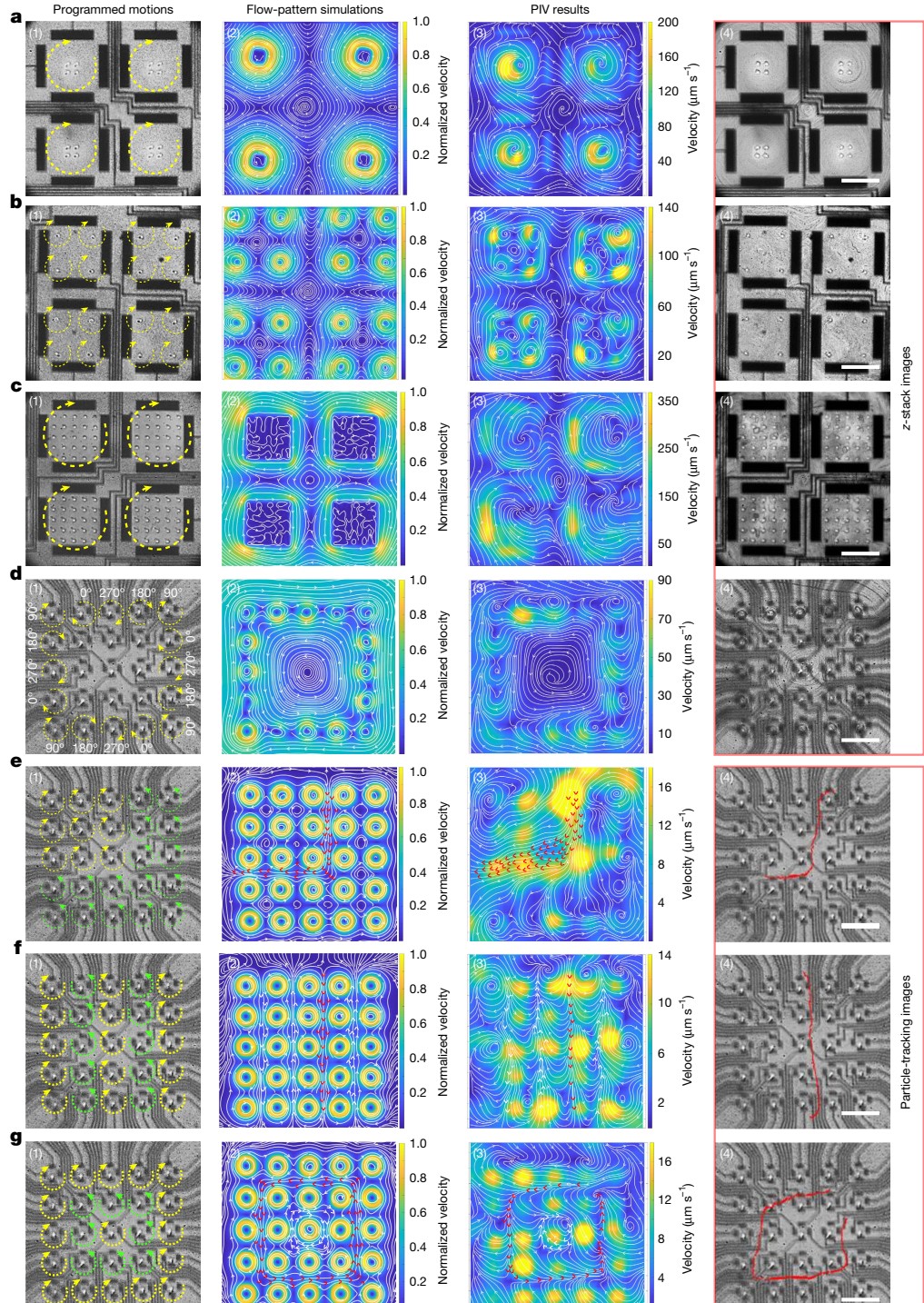

**Fig. 4 | Dynamic fluid manipulation by the gel microcilia arrays. a–c,** Fluid control achieved by varying the spatial arrangement and density of gel microcilia within an actuation cell (rotational bending frequency = 40 Hz). **a,** (1) Programmed motion of a 4 × 4 array with four cilia centred in each cell, all rotating clockwise. **a,** (2) 2D simulation shows four clockwise vortices and one inter-cell anticlockwise vortex. **b,** (1) 4 × 4 array with four corner cilia per cell, all rotating clockwise. **b,** (2) Each cell produces five vortices (four clockwise, one central anticlockwise), forming inter-cell anticlockwise flows. **c,** (1) 10 × 10 array with 5 × 5 densely packed cilia per cell, all rotating clockwise. **c,** (2) Hydrodynamic interference cancels intra-cell vortices, leaving peripheral clockwise and central anticlockwise vortices. **d–g,** Dynamic reprogramming of cilia motion for flow control (20 Hz). **d,** (1) 5 × 5 individually controlled array with outer 16 cilia generating metachronal waves; inner cilia stationary. **d,** (2) Simulation shows a central anticlockwise vortex. **e,** (1) 5 × 5 array with a 3 × 3 top-left subarray rotating

clockwise, others anticlockwise. **e,** (2) Clockwise L-shaped flow forms between these regions. **f,** (1) Alternating column actuation (columns 1, 3 and 5 clockwise; columns 2 and 4 anticlockwise). **f,** (2) Downward flows between columns 1–2 and 3–4 and upward flows between columns 2–3 and 4–5. **g,** (1) 5 × 5 array with outer and central cilia rotating clockwise, middle ring anticlockwise. **g,** (2) Anticlockwise flow between the outer and middle rings and clockwise flow between the middle ring and the central cilium. Experimental PIV flow fields (**a** (3)–**g** (3)), z-stack particle traces (**a** (4)–**d** (4)) and particle-trajectory tracking (**e** (4)–**g** (4)) all align with simulations **a** (2)–**g** (2). Red arrows in **e**–**g** highlight that flow directions are consistent with the particle-trajectory-tracking results. The gel microcilia actuators used in the flow experiments have a diameter of 10 μm and a height of 90 μm. The working solution is 0.00769 mol l⁻¹ NaCl. Scale bars, 200 μm (**a** (4)–**c** (4)); 150 μm (**d** (4)–**g** (4)).

regulation of embryonic development[8]. As shown in Fig. 4, similarly, our hydrogel cilia enable controllable fluid transport and directional particle motion (Fig. 4e–g).

We show two strategies for microscale fluid manipulation control. First, we manipulate fluid flow through spatial and density configurations of gel microcilia within one actuation cell under identical clockwise actuation signals (Fig. 4a–c). Experiments 1–3 show that cilia position and density can govern the vortex pattern and direction under the synchronized clockwise motion. In experiment 1, four gel microcilia positioned at the centre of each actuation cell generate clockwise vortices that induce an anticlockwise vortex in the inter-cell region (Fig. 4a and Supplementary Video 24). Experiment 2 uses a corner-positioned configuration; each actuation cell produces five vortices (four clockwise and one central anticlockwise), with neighbouring vortices inducing an inter-cell anticlockwise flow (Fig. 4b and Supplementary Video 25). To further investigate the dense gel microcilia array motion for fluid pumping, experiment 3 builds 25 gel microcilia actuators within one actuation cell. The results indicate that tight cilia spacing cancels intra-cell vortices owing to hydrodynamic interference between neighbouring cilia, whereas peripheral clockwise vortices and a central anticlockwise vortex between cells emerge (Fig. 4c and Supplementary Video 26). Particle image velocimetry (PIV) (Fig. 4a–c (3)) and z-axis stack images of particles traces (Fig. 4a–c (4)) confirm simulated flow patterns (Fig. 4a–c (2)) across all configurations.

Second, we demonstrate dynamic reprogramming of flow patterns through individually addressable gel microcilia (Fig. 4d–g). Experiments 4–7 demonstrate that reprogrammable coordinated motions can control the flow structure and direction in a versatile way. In experiment 4, 16 gel microcilia in the outermost ring rotate clockwise with a 90° phase difference between adjacent actuators to form metachronal waves, whereas the remaining cilia remain stationary. This configuration produces a centralized anticlockwise vortex (Fig. 4d and Supplementary Video 27). In experiment 5, a region-specific actuation strategy is used: a subarray of 3 × 3 gel microcilia in the top-left corner rotates clockwise, whereas the remaining gel microcilia rotate anticlockwise, generating clockwise L-shaped flows (Fig. 4e and Supplementary Video 28). Experiment 6 implements columnar alternating motion: gel microcilia in columns 1, 3 and 5 (15 gel microcilia) rotate clockwise and those in columns 2 and 4 (10 gel microcilia) rotate anticlockwise, resulting in bidirectional vertical flows. Downward flows dominate between columns 1 and 2 as well as between columns 3 and 4, whereas upward flows occur between columns 2 and 3 and between columns 4 and 5 (Fig. 4f and Supplementary Video 29). Experiment 7 uses concentric ring actuation: the outermost ring (16 gel microcilia) and the central cilium rotate clockwise, whereas the middle ring (8 gel microcilia) rotates anticlockwise, establishing nested vortices—anticlockwise flow between the outer and middle rings and clockwise flow between the middle ring and central cilium (Fig. 4g and Supplementary Video 30). PIV data (Fig. 4d–g (3)), a z-stack image of particles traces (Fig. 4d (4)) and particle trajectory tracking (Fig. 4e–g (4)) all align with simulations (Fig. 4d–g (2)), confirming the flow directionality and patterns across the working area. All of the dynamic fluid simulation results are shown in Supplementary Video 31. From the PIV measurement, we estimate that the maximum flow velocities generated by cilia in cases in Fig. 4a–d range between 50 and 250 μm s$^{-1}$. In the cases in Fig. 4e–g, particle-tracking measurements indicate velocities of approximately 55 μm s$^{-1}$, 18 μm s$^{-1}$ and 22 μm s$^{-1}$, respectively.

## Discussion

Present hydrogel actuation performance declines in complex ionic environments, that is, solutions with several dominant ions. Performance decreases from DI water to physiological saline and DPBS and is expected to decline further in biological contexts. Potential solutions include optimizing the hydrogel monomer composition (for example, incorporating 2-acrylamido-2-methyl-1-propanesulfonic acid for enhanced hydrogel network polarizability) and reducing cilium diameter (≤2 μm) to lower the ion redistribution time for enhanced performance. Furthermore, enhancing electric-field uniformity and intensity by means of 3D microelectrodes or reducing electrode spacing (Extended Data Fig. 7c) could improve performance. Although reduced electrode spacing may affect neighbouring electric fields, our experiments indicate negligible interference at a spacing of 5 μm between actuation cells (Fig. 3a); further investigation is required in the future.

We predict that continued improvements in gel microcilia performance, combined with advanced microfabrication/nanofabrication and flexible electronics fabrication technologies, will enable their broad applications across many fields.

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

## Methods

### Single-layer microelectrodes fabrication

Extended Data Fig. 2 illustrates the fabrication process of single-layer microelectrodes. Below are the step-by-step details:

Step 1. Preparation of the polyimide substrate (Extended Data Fig. 2a). This work uses PI2611 (HD MicroSystems) as the backbone material. PI2611 is poured onto a blank glass substrate and then spin-coated at 1,500 rpm for 30 s. The glass substrate used to hold the printed hydrogel microactuators must have a thickness of less than 300 μm, as this is the working range of the TPP laser. For this purpose, we use a 180-μm-thick glass substrate for hydrogel printing. No specific thickness is required for the glass substrate used for moulded hydrogel fabrication.

Step 2. Curing the PI2611 polyimide substrate (Extended Data Fig. 2b). We place the spin-coated substrate on a hotplate and heat it from room temperature to 150 °C. We hold this temperature for 10 min and then increase it to 200 °C. The temperature ramp rate is 20 °C per minute. This temperature is maintained for 5 h to fully cure the polyimide.

Step 3. Photoresist coating (Extended Data Fig. 2c). We pour the positive photoresist AZ ECI 3012 (MicroChemicals GmbH) onto the polyimide substrate and spin-coat the photoresist for 30 s at 5,000 rpm.

Step 4. Soft baking (Extended Data Fig. 2d). We bake the positive photoresist at 90 °C for 90 s.

Step 5. Ultraviolet exposure (Extended Data Fig. 2e). We expose the substrate for 8 s using the MJB4 mask aligner (SUSS MicroTec). The ultraviolet density of this machine is 14.3 mJ cm$^{-2}$ and the patterns are defined using a photomask.

Step 6. Post-exposure bake (Extended Data Fig. 2f). We bake the exposed substrate at 110 °C for 90 s.

Step 7. Development (Extended Data Fig. 2g). We develop the substrate in AZ 726 (MicroChemicals GmbH) developer for 60 s to reveal the patterns.

Step 8. Platinum sputtering (Extended Data Fig. 2h). We deposit a 150-nm-thick platinum (Pt) layer onto the substrate using sputtering.

Step 9. Lift-off process (Extended Data Fig. 2i). We dip the substrate in TechniStrip Micro D350 (MicroChemicals GmbH) photoresist stripper to remove unwanted material and obtain the final microelectrode structures.

### Hydrogel solutions for 3D printing

Four hydrogel precursor solutions are prepared for TPP-based 3D printing. Acrylic acid (AAc, Sigma-Aldrich) and acrylamide (AAm, Sigma-Aldrich) served as monomers, $N,N'$-methylenebisacrylamide (BIS, Sigma-Aldrich) as the crosslinker and Omnirad 2100 (IGM Resins) as the photoinitiator. Ethylene glycol was used as the base solvent.

The four solutions differ in the mass fractions of AAc, AAm and the photoinitiator. The molar masses of AAc (72.06 g mol$^{-1}$) and AAm (71.08 g mol$^{-1}$) are nearly identical; therefore, maintaining a constant total mass fraction of AAc and AAm effectively corresponds to a constant overall molar concentration of monomers. This ensures that, across all formulations, the total monomer concentration remains comparable when only the relative ratio of AAc to AAm varies.

Because the printability of the precursor solution depends on the AAc-to-AAm ratio, the photoinitiator fraction is experimentally adjusted to achieve consistent printing quality. For each monomer ratio, several photoinitiator concentrations are tested under identical printing parameters and the optimal concentration is determined as the one that produced structures without overexposure or underexposure. Overexposure and underexposure are defined as follows: when printing a 10-μm feature under identical parameters, a fabricated structure larger than 10 μm is considered overexposed, whereas one smaller than 10 μm is considered underexposed. All mass fractions reported below are calculated with respect to the initial mass of ethylene glycol before the addition of monomers, crosslinker or photoinitiator.

Solution 1: AAc 15 wt%, AAm 60 wt%, BIS 2 wt%, photoinitiator 7.5 wt%.

Solution 2: AAc 30 wt%, AAm 45 wt%, BIS 2 wt%, photoinitiator 18 wt%.
Solution 3: AAc 45 wt%, AAm 30 wt%, BIS 2 wt%, photoinitiator 28 wt%.
Solution 4: AAc 60 wt%, AAm 15 wt%, BIS 2 wt%, photoinitiator 38 wt%.

### Hydrogel microactuator 3D printing

A commercial TPP system (Photonic Professional GT, Nanoscribe GmbH) is used to fabricate hydrogel microactuators. The printing is performed using a 25× objective lens with an exposure power of 15 mW. The slicing and hatching distances are set to 300 nm and 200 nm, respectively, with a 45° hatching angle between adjacent layers (Extended Data Fig. 3).

We immerse the printed sample in an ethylene glycol bath to develop the hydrogel microactuator structure. Subsequently, the sample is transferred to a DI water bath for 10 min. We repeat this process three times to fully replace the solvent inside the hydrogel from ethylene glycol to DI water. Afterwards, the aqueous environment can be adjusted for different experiments accordingly.

### Comparison between the centimetre–millimetre hydrogel and the micrometre hydrogel

In contrast to the fast bending mechanism of the micrometre hydrogel actuator, the internal ion migration is much slower in previously reported centimetre–millimetre-scale ionic hydrogels not fabricated by TPP[49,50]. Even at the same electric-field intensity as our gel microcilia system, for example, 10,000 V m$^{-1}$, the H$^+$ ions and Na$^+$ ions would take 0.3 s and 16.7 s, respectively, to traverse a 1-cm distance, which is a typical thickness of a centimetre-scale hydrogel actuator. This ion migration time is orders of magnitude slower than at the micrometre scale. Moreover, its larger pore size (Extended Data Fig. 1b (i)) and smaller effective EDL surface area reduce ion transport, thereby weakening bending. Finally, achieving such high electric fields in large systems needs impractical voltages (for example, 200 V for 2-cm-spaced electrodes), which can trigger electrolysis and other vigorous electrochemical reactions.

In fact, the internal ion migration in millimetre-scale hydrogel is trivial compared with ion partitioning at the gel–liquid interface. Therefore, previously reported millimetre-scale hydrogels[49,50] operate through mechanisms distinct from those proposed in the present work. They bend by means of bath-induced pH gradients or interfacial osmotic effects, driven by electrolysis or ion partitioning across the gel–solution interface[42–44]. By contrast, micrometre-scale hydrogels depend on internal ion migration and nanometre-scale channels, resulting in opposite bending directions and response times more than two orders of magnitude faster (Extended Data Fig. 5 and Supplementary Videos 6 and 7). For example, in DI water, a millimetre-scale actuator bends towards the anode in about 120 s, whereas a micrometre-scale actuator bends towards the cathode in about 0.2 s (Extended Data Fig. 5a,b). In 0.15380 mol l$^{-1}$ NaCl, the millimetre-scale actuator bends towards the cathode in about 30 s, whereas the micrometre-scale actuator bends towards the anode in about 0.3 s (Extended Data Fig. 5c,d).

### Mechanism in DI water

For the millimetre-scale mechanism in DI water (Extended Data Fig. 5a), under the applied voltage, electrolysis produces H$^+$ near the anode and OH$^-$ near the cathode. Because the AAc-co-AAm network is pH-sensitive, acidic conditions convert –COO$^-$ to –COOH (reducing repulsion force and causing contraction), whereas alkaline conditions convert –COOH to –COO$^-$ (increasing repulsion and causing swelling). The bath-induced pH gradient therefore shrinks the anode-side region and swells the cathode-side region, resulting in bending towards the anode[42].

By contrast, for the micrometre-scale mechanism in DI water (Extended Data Fig. 5b), the micrometre-scale hydrogel bends towards the cathode. Here fixed –COOH groups partially dissociate into –COO$^-$ and mobile H$^+$. To maintain electroneutrality, the dissociated H$^+$ ions

largely remain confined within the hydrogel. Under an external field, these $H^+$ ions migrate and accumulate on the cathode-facing side, locally lowering pH and inducing network contraction, which bends the hydrogel towards the cathode.

## Mechanism in 0.15380 mol l$^{-1}$ NaCl

For the millimetre-scale mechanism in 0.15380 mol l$^{-1}$ NaCl (Extended Data Fig. 5c), in a saline environment, the external field drives the migration of all mobile ions in the bath. The fixed negative charges of the hydrogel influence ion partitioning at the gel–solution interface[42–45], producing non-uniform ion concentrations across four regions (gel anode side: region 2; gel cathode side: region 1; solution anode side: region 4; solution cathode side: region 3). According to Flory's theory[51], the local osmotic pressure difference is $\Delta\pi = RT\sum_i (c_{ig} - c_{is})$, in which $\Delta\pi$ is the pressure difference, $c_{ig}$ is the ion concentration in the hydrogel, $c_{is}$ is the ion concentration in solution, $R$ is the gas constant and $T$ is temperature. Under steady-state conditions, the osmotic pressure difference on the anode side $\Delta\pi_{\text{Anode side}} = RT\sum_i (c_{i\text{region2}} - c_{i\text{region4}})$ exceeds that on the cathode side $\Delta\pi_{\text{Cathode side}} = RT\sum_i (c_{i\text{region1}} - c_{i\text{region3}})$. This imbalance causes the anode side to swell more, bending the hydrogel towards the cathode.

At the microscale, the dominant factor is the ion concentration gradient inside the hydrogel rather than at the interface. The relevant osmotic pressure is $\Delta\pi_{\text{Inside the hydrogel}} = RT\sum_i (c_{i\text{region1}} - c_{i\text{region2}})$, which remains positive, making the cathode side swell more and driving bending towards the anode (Extended Data Fig. 5d).

## Osmotic pressure analysis

Previously reported centimetre-scale and millimetre-scale hydrogel actuators operated in enclosed solution environments, in which the hydrogel was immersed in a bath equipped with two electrodes on the sidewalls[44,50,52] (Extended Data Fig. 6a). In such systems, most ions in the bath are consumed in establishing concentration gradients across the hydrogel–solution interfaces, generating osmotic pressure differences that drive the hydrogel to bend towards the cathode[42–44].

By contrast, our microscale hydrogel cilia operate within a localized region (200 μm × 200 μm × 90 μm) inside a much larger bath (4 cm × 4 cm × 3 mm) (Extended Data Fig. 6b). The large bath volume serves as an ion reservoir, allowing rapid diffusion of external ions into the actuation region (Extended Data Fig. 6b (ii)) and thereby preventing the formation of substantial ion concentration gradients or osmotic pressure differences across the hydrogel–solution interfaces.

To further validate this reason, osmotic-pressure simulations are performed under two distinct scenarios. Here osmotic pressure refers to the interfacial osmotic-pressure difference that governs centimetre-scale and millimetre-scale hydrogel actuation (as defined in Extended Data Fig. 5c). Direct comparison between simulations at different physical scales is not meaningful because the results would not be dimensionally consistent. Therefore, both simulations are conducted at the micrometre scale, with the only variable being the presence or absence of ion sources at the boundaries.

Case 1: macroscale actuation mimicked at the microscale (Extended Data Fig. 6c (i)). The simulation domain is a 2D region of 200 μm × 150 μm containing a hydrogel of dimensions 10 μm × 90 μm, reproducing the relative size ratio between the hydrogel and the bath in the centimetre–millimetre-scale experiments. No external ion source is applied, corresponding to the enclosed environment characteristic of macroscale systems.

Case 2: microscale actuation with ion exchange (Extended Data Fig. 6c (ii)). The geometry and parameters are identical to case 1, except that ion-source boundary conditions are imposed on three sides of the simulation domain, allowing continuous ion exchange between the system and the surroundings.

As shown in Extended Data Fig. 6d, introducing ion sources at the boundaries markedly reduced the osmotic-pressure difference across the hydrogel–solution interface. This result confirms that, compared with the macroscale condition, the osmotic contribution is greatly diminished at the microscale and is no longer the dominant mechanism governing hydrogel actuation.

## Coupled electro-chemo-mechanical model

A fully coupled electro-chemo-mechanical model is developed here. The model simultaneously resolves: (1) ionic concentration distribution under an external electric field and the fixed charges of the hydrogel network; (2) the resulting osmotic body forces generated by non-uniform ion distributions; and (3) the gel deformation driven by these body forces. At this stage, the model focuses on explaining the deformation of hydrogels in ionic solutions under external electric fields and does not include fluid–structure interactions between the gel and the surrounding liquid.

## Ionic concentration distribution

The Nernst–Planck equation describes the ion concentration in a charged hydrogel network environment:

$$J_\alpha = -D_\alpha\nabla c_\alpha - z_\alpha\mu_\alpha c_\alpha\nabla\varnothing + c_\alpha v \tag{1}$$

in which $J_\alpha$ is the ion flux (mol (m$^2$ s)$^{-1}$), $D_\alpha$ is the diffusion coefficient of ion species $\alpha$, $c_\alpha$ is the concentration of ion species $\alpha$, $z_\alpha$ is the charge number of the ion (valence), $\mu_\alpha$ is the mobility of the ion, $\varnothing$ is the electric potential and $v$ is the velocity of the fluid.

The change in ion concentration $c_\alpha$ over time is governed by the continuity equation, which states that the change rate of the ion concentration is equal to the net flux of ions plus any sources or sinks (chemical reactions, for instance). It is expressed as:

$$\frac{\partial c_\alpha}{\partial t} + \nabla\cdot J_\alpha = r_\alpha(c_\beta) \tag{2}$$

in which $\frac{\partial c_\alpha}{\partial t}$ is the ion concentration change rate, $\nabla\cdot J_\alpha$ represents the net flow of ions into or out of a region and $r_\alpha(c_\beta)$ is a source term representing the creation or consumption of ion $\alpha$ owing to chemical reactions or other processes involving species $\beta$.

By substituting the Nernst–Planck equation (equation (1)) into the continuity equation (equation (2)), we obtain:

$$\frac{\partial c_\alpha}{\partial t} = \nabla\cdot[D_\alpha\nabla c_\alpha + z_\alpha\mu_\alpha c_\alpha\nabla\varnothing - c_\alpha v] + r_\alpha(c_\beta) \tag{3}$$

The Poisson equation relates the electric potential distribution to the charge density in the system. It is given by

$$\nabla^2\varnothing = -\frac{\rho}{\in} \tag{4}$$

in which $\rho$ is the charge density (total charge per unit volume) and $\varepsilon$ is the permittivity of the medium, expressed as $\varepsilon_r\varepsilon_0$, in which $\varepsilon_0$ is the vacuum permittivity and $\varepsilon_r$ is the relative permittivity of the medium.

In this work, the AAc-co-AAm hydrogel network contains –COOH groups, whose ionization equilibrium is influenced by $H^+$ migration. This equilibrium affects the network charge and mobility of other ions. This equilibrium is included in the modelling by

$$\frac{C_{\text{R-COO}^-}\cdot C_{\text{H}^+}}{C_{\text{R-COOH}} - C_{\text{R-COO}^-}} = K_a = 5.6\times 10^{-5} \tag{5}$$

in which $C_{\text{R-COO}^-}$ is the concentration of the dissociated function group in the hydrogel, $C_{\text{H}^+}$ is the $H^+$ ion concentration in the hydrogel and $C_{\text{R-COOH}}$ is the concentration of the carboxyl group in the hydrogel. The value of $C_{\text{R-COOH}}$ is obtained from the initial hydrogel solution. Equations (3)–(5) collectively govern the ion concentration distribution.

## Forces generated by non-uniform ion distributions

The force induced by ionic distributions can be separated into two contributions. The first contribution comes from the H⁺ effect (pH-dependent). H⁺ modulate the dissociation equilibrium of –COO⁻ groups within the hydrogel network. Variations in the fixed –COO⁻ concentration govern network swelling or contraction. The corresponding force can be expressed as:

$$f_{pH} = E(-\nabla(FC_{R\text{-}COO^-}))$$ (6)

in which $F$ is the Faraday constant and $E$ is the local electric field, which can be obtained from

$$E = -\nabla\varnothing.$$ (7)

The second contribution comes from the local pressure difference induced by ion concentration. According to Flory's theory[51], the local pressure relates to the ion concentration:

$$\pi = RT\sum_i c_i$$ (8)

in which $\pi$ is the local pressure, $c_i$ is the ion concentration, $R$ is the gas constant and $T$ is temperature.

The force generated by the non-uniform ion concentration can be expressed by the negative gradient of the local pressure as

$$f = -\nabla\pi = -\nabla\left(RT\sum_i c_i\right).$$ (9)

The local electric potential $\varnothing$, function group concentration $C_{R\text{-}COO^-}$ and ion concentration $c_i$ can be calculated from equations (3)–(5).

## Mechanical deformation

The deformation of the hydrogel was modelled as a nonlinear hyperelastic material considering geometric nonlinearity. To describe its constitutive behaviour, the first-order compressible Ogden model was used. The strain energy density function is expressed as

$$W = \frac{\mu}{\alpha}(\lambda_1^\alpha + \lambda_2^\alpha + \lambda_3^\alpha - 3) + \frac{1}{D}(J-1)^2$$ (10)

in which $\lambda_1$, $\lambda_2$ and $\lambda_3$ are the principal stretch ratios, $J = \lambda_1\lambda_2\lambda_3$ is the volume ratio, $\mu$ and $\alpha$ are material constants and $D$ is a compressibility parameter related to the bulk modulus $K = 2/D$. The Poisson's ratio $v = 0.42$ is chosen to define the degree of compressibility. The material parameters ($\mu$, $\alpha$) are determined by fitting the Ogden model to the data obtained from atomic force microscopy tests and are listed in Extended Data Table 2.

## PIV analysis

The 2D in-plane velocity is calculated using the open-source software PIVlab[53]. Background subtraction is applied to the particle images to enhance image quality. The multipass fast Fourier transform window deformation algorithm is used to improve the accuracy of displacement estimation. The interrogation area is initially set to 64 × 64 pixels with a step size of 32 pixels, corresponding to a 50% overlap between adjacent interrogation windows. In the second pass, the interrogation area is reduced to 32 × 32 pixels with a step size of 16 pixels. For higher precision, sub-pixel displacements are estimated using a Gaussian 2 × 3 point estimator. The PIV flow pattern results are shown in Fig. 4.

It should be noted that the patterned electrodes on the substrate influence the accuracy of PIV analysis. For cases in Fig. 4a–c, in which the electrode density is relatively low, the PIV results are reliable. For case in Fig. 4d–g, the electrode density is higher and the trajectories of the PIV tracer particles overlap substantially with the underlying electrodes, leading to larger errors in velocity quantification.

## 2D flow simulations

For the numerical simulation of the interaction between the cilia array and fluid, we use the hybrid finite difference/finite element immersed boundary method[54], implemented in the open-source software IBAMR, a widely tested C++ framework for the immersed boundary method. The immersed boundary formulation of the problem describes the momentum and velocity of the coupled fluid–structure system in Eulerian form, whereas the deformation and elastic response of the immersed structure is in the Lagrangian form. This study uses 2D simulations, as they effectively capture the flow patterns. Let $\mathbf{x} = (x_1, x_2) \in \Omega \subset \mathbb{R}^2$ denote Cartesian physical coordinates, in which $\Omega$ represents the physical region that is occupied by the coupled fluid–structure system, let $\mathbf{X} = (X_1, X_2) \in W \subset \mathbb{R}^2$ denote Lagrangian material coordinates that are attached to the structure, in which $W$ is the Lagrangian domain, and let $\chi(\mathbf{X}, t) \in \Omega$ denote the physical position of material point $\mathbf{X}$ at time $t$. The strong form of the equations of motion is:

$$\rho\frac{Du}{Dt}(\mathbf{x}, t) = -\nabla p(\mathbf{x}, t) + \mu\nabla^2\mathbf{u}(\mathbf{x}, t) + \mathbf{f}^c(\mathbf{x}, t)$$
$$\nabla \cdot \mathbf{u}(\mathbf{x}, t) = 0$$ (11)

$$\mathbf{f}^c(\mathbf{x}, t) = \int_U \nabla_\mathbf{X} \cdot \mathbb{P}^e(\mathbf{X}, t)\delta(\mathbf{x} - \chi(\mathbf{X}, t))\mathbf{f}^e(\mathbf{x}, t)d\mathbf{X}$$
$$- \int_{\partial U} \mathbb{P}^e(\mathbf{X}, t)\mathbf{N}(\mathbf{X})\delta(\mathbf{x} - \chi(\mathbf{X}, t))\mathbf{f}^e(\mathbf{x}, t)d\mathbf{X}$$ (12)
$$\frac{\partial\chi}{\partial t}(\mathbf{X}, t) = \int_\Omega \mathbf{u}(\mathbf{x}, t)\delta(\mathbf{x} - \chi(\mathbf{X}, t))d\mathbf{X}$$

in which $\rho$ is the mass density, $\mathbf{u}(\mathbf{x}, t)$ is the Eulerian velocity field, $\mu$ is the dynamic viscosity, $\mathbf{f}^e(\mathbf{x}, t)$ is the Eulerian elastic force density, $\mathbb{P}^e(\mathbf{X}, t)$ is the first Piola–Kirchhoff elastic stress tensor, $\delta(\mathbf{x})$ is the 2D delta function and $\mathbf{N}(\mathbf{X})$ is the normal vector along the fluid–solid interface. In the computations of this study, the physical domain is $\Omega = [-L, L]$ $[-L, L]$, in which $L$ is 600 μm for simulations corresponding to fluid experiments 1–3 and 400 μm for fluid experiments 4–7. A zero-gradient boundary condition is applied to boundaries. A staggered-grid finite difference scheme is used to discretize the incompressible Navier–Stokes equations in space. The spatial resolution is $\Delta x = L/128$ for simulations 1–3 and $\Delta x = L/64$ for simulations 4–7. The total number of the Cartesian grid is $\mathcal{O}(10^5)$. The circular section of the cilia is discretized into a mesh of triangular elements with an average node spacing of $\Delta X = L/128$. Time-stepping is performed using an implicit scheme proposed by Newren et al.[55]. The time step size is adjusted to satisfy the Courant–Friedrichs–Lewy condition, with a stability number of approximately 0.1.

In these 2D flow-field simulations, solid spheres with a diameter of 10 μm are used to approximate the hydrogel cilia. Under the prescribed motions, interactions between the sphere edges and the surrounding fluid generated relatively high flow velocities. However, in reality, hydrogel cilia are porous and soft materials and their interaction with the fluid under identical motions does not produce flow velocities of the same magnitude as in the simulations. The primary aim of the simulation is to qualitatively predict flow-field patterns, not to precisely quantify flow velocities, so the simulated flow patterns are shown in this work (Fig. 4a–g (2)).

## Data availability

The biological cilia data are reproduced from the work of Okada et al.[8]. All data that support the findings of this study are provided in the main text, methods and the Supplementary Information.

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

**Acknowledgements** We thank W. Kang, M. Zhang and X. Lyu for their assistance with the experiments. This work is supported by the Max Planck Society, European Research Council (ERC) Advanced Grant SoMMoR project with grant no. 834531, German Research Foundation (DFG) Soft Material Robotic Systems (SPP 2100) Priority Programme with grant no. 2197/3-1, Max Planck Queensland Centre for the Materials Science of Extracellular Matrices and Hong Kong University of Science and Technology Startup Fund with grant no. R9910.

**Author contributions** Z.L., W.H. and M.S. proposed and designed the research. Z.L. performed the experiments and analysed the data, with the help of Z.R., Chunxiang Wang, W.W., J.K., C.H., X.C. and H.W. Z.L. and Che Wang developed the fully coupled model. Z.L. and W.W. performed the fluid-flow simulations. Z.L. and S.S. built the control circuit. M.S. and W.H. supervised the research. The manuscript was written by Z.L., W.H. and M.S., with input from all authors. All authors discussed the results and commented on or edited the manuscript.

**Funding** Open access funding provided by Max Planck Society.

**Competing interests** Max Planck Innovation filed a patent application on behalf of Z.L., W.H. and M.S. based on the methods and results presented here (application no. EP25215159.2). The other authors declare no competing interests.

**Additional information**
**Correspondence and requests for materials** should be addressed to Wenqi Hu or Metin Sitti.

**a** (i) Conventional flood-exposure fabrication

(ii) TPP printing fabrication

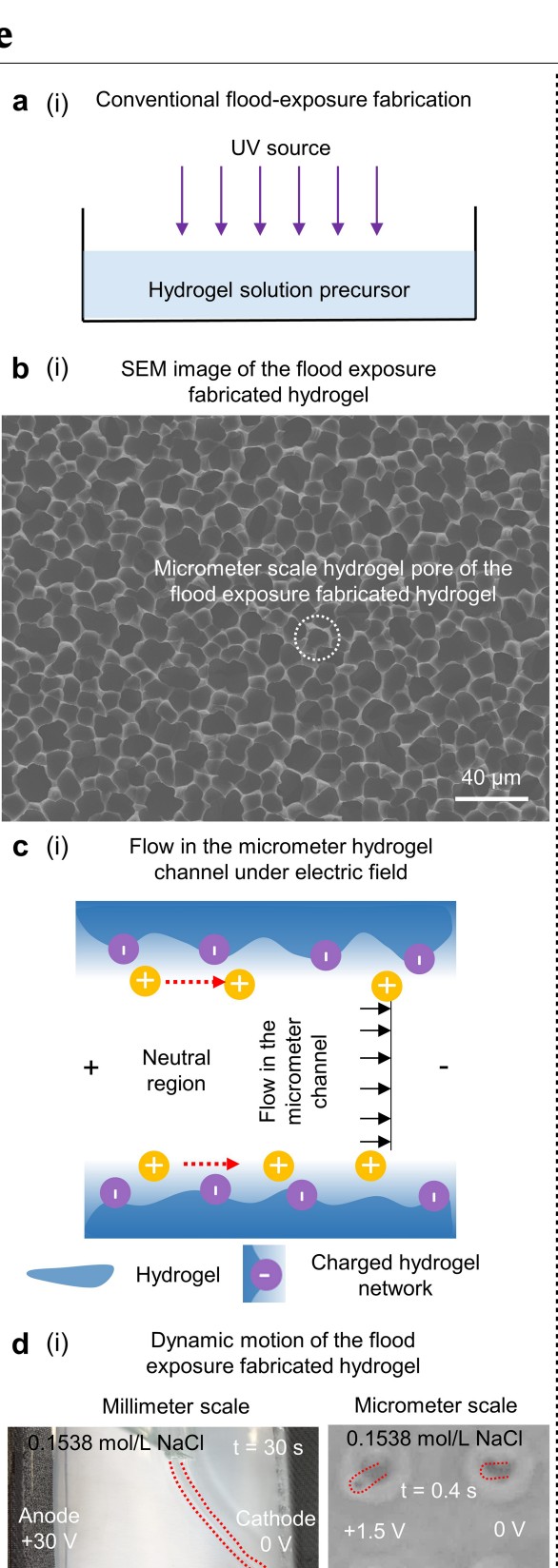

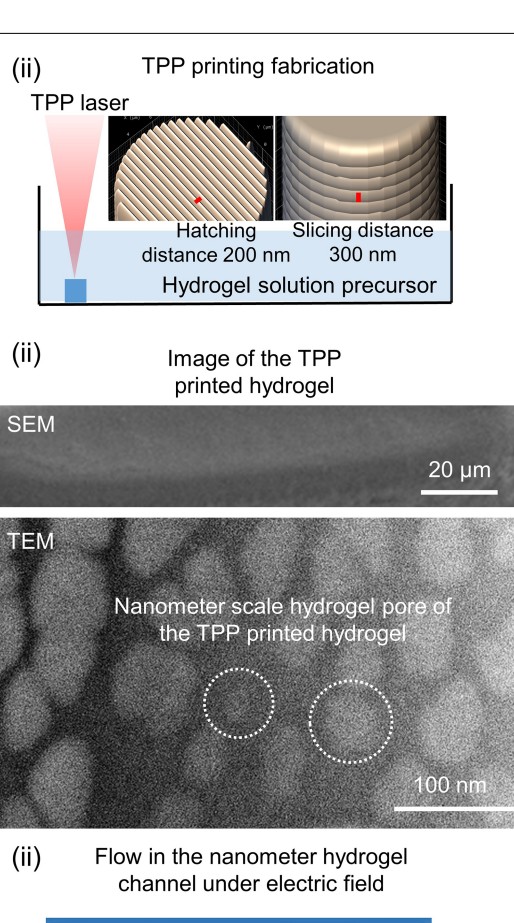

**b** (i) SEM image of the flood exposure fabricated hydrogel

(ii) Image of the TPP printed hydrogel

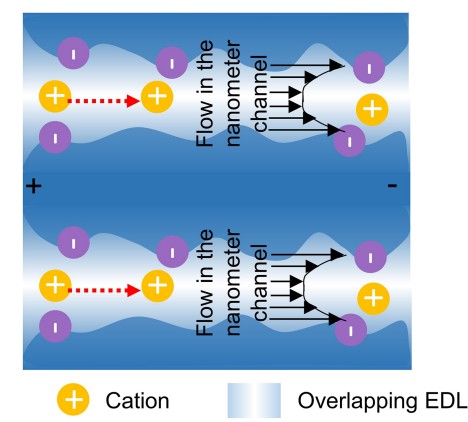

**c** (i) Flow in the micrometer hydrogel channel under electric field

(ii) Flow in the nanometer hydrogel channel under electric field

**d** (i) Dynamic motion of the flood exposure fabricated hydrogel

(ii) Dynamic motion of a TPP printed micrometer scale hydrogel

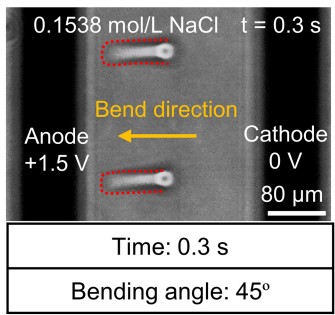

| Time: 30 s | Time: 0.4 s |
|---|---|
| Bending Angle: 40° | Bending angle: 10° |

| Time: 0.3 s |
|---|
| Bending angle: 45° |

**Extended Data Fig. 1** | See next page for caption.

**Extended Data Fig. 1 | Comparison between hydrogels fabricated by conventional flood exposure and TPP. a**, (i) Schematic of the conventional flood-exposure fabrication process. (ii) Schematic of the TPP fabrication process, which allows tuning of printing by adjusting parameters such as hatching and slicing. **b**, (i) SEM image of a hydrogel fabricated by flood exposure, showing pore sizes in the range 10–30 μm. (ii) SEM and TEM images of the TPP-printed hydrogel. At this magnification, the SEM image shows a nearly featureless surface without discernible pores. The TEM image shows pore sizes ranging from 20 to 80 nm. **c**, Schematics illustrating the influence of pore size on electro-osmotic flow. (i) Flow in a micrometre-scale ionic hydrogel channel. (ii) Enhanced flow in a nanometre-scale ionic hydrogel channel. **d**, Dynamic performance of hydrogels fabricated by different methods. (i) Millimetre-scale (left) and micrometre-scale (right) hydrogels fabricated by flood exposure. (ii) Micrometre-scale hydrogel fabricated by TPP printing.

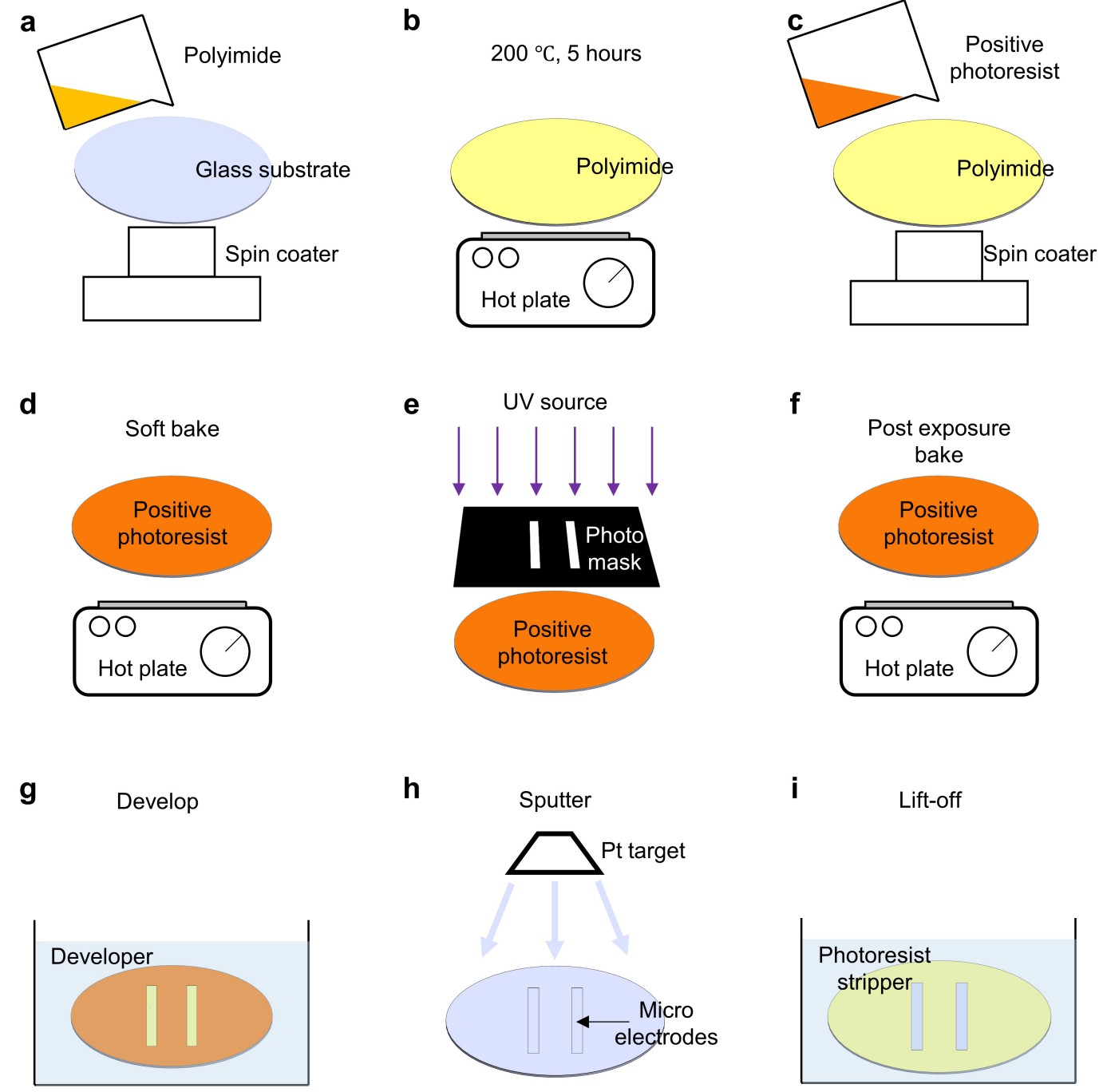

**Extended Data Fig. 2 | Single-layer microelectrodes fabrication process.** The microelectrodes are fabricated using a conventional lift-off method. A detailed step-by-step procedure is provided in Methods.

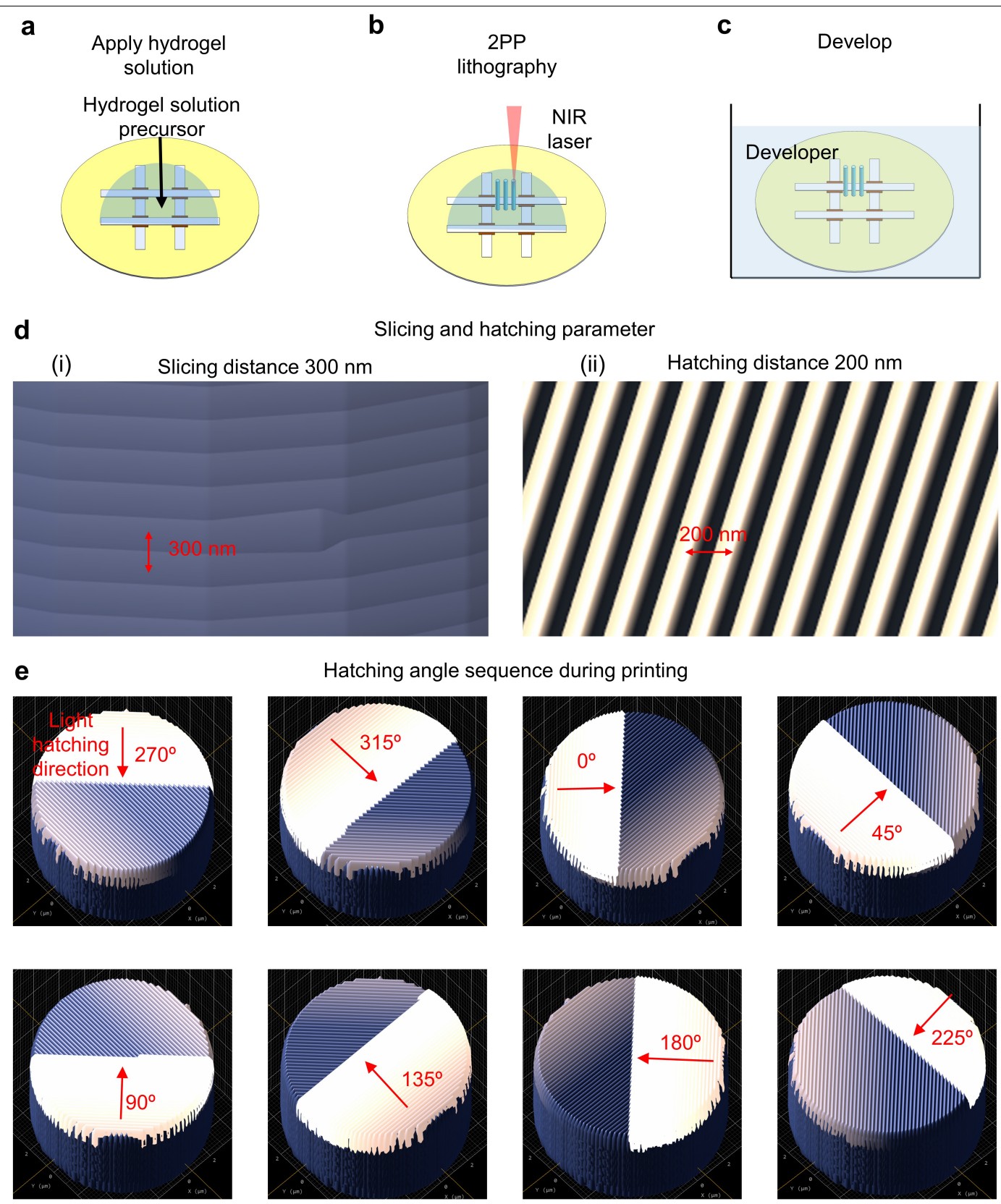

**a** Apply hydrogel solution

Hydrogel solution precursor

**b** 2PP lithography

NIR laser

**c** Develop

Developer

**d** Slicing and hatching parameter

(i) Slicing distance 300 nm

300 nm

(ii) Hatching distance 200 nm

200 nm

**e** Hatching angle sequence during printing

Light hatching direction

270°

315°

0°

45°

90°

135°

180°

225°

**Extended Data Fig. 3 | Hydrogel microactuator printing. a–c**, The printing process of the hydrogel microactuator. **d**, Slicing and hatching parameters used during TPP printing. **e**, In this work, the hatching angle between successive layers was offset by 45°, with anticlockwise rotation, to ensure uniformity of the printed structures.

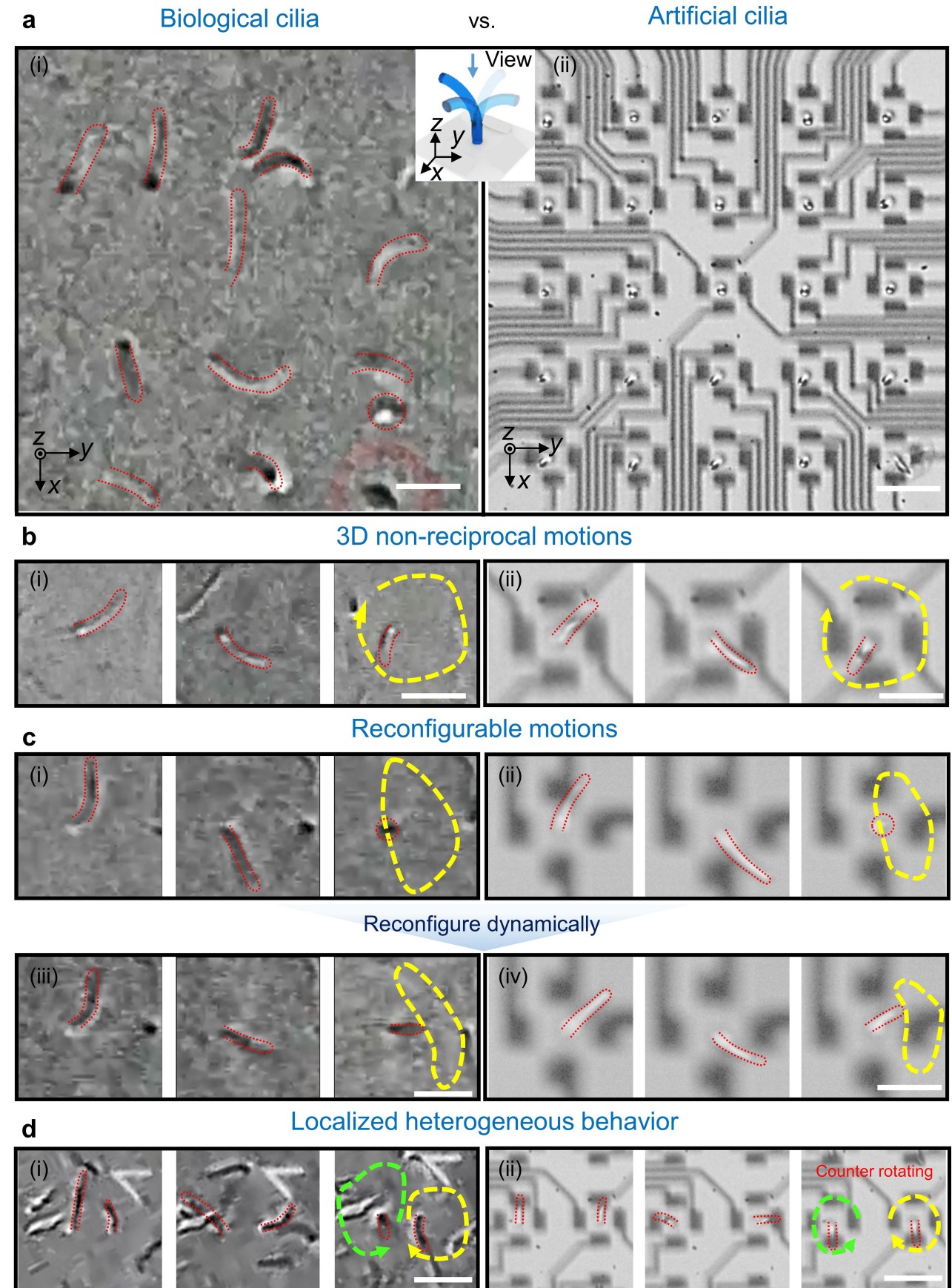

**Extended Data Fig. 4** | See next page for caption.

**Extended Data Fig. 4 | Comparison between biological cilia and artificial hydrogel cilia.** The left column shows mouse ventral nodal cilia[8] (with permission from the publisher, Elsevier) and the right column shows the hydrogel cilia developed in this work. **a**, Cilia arrays (Supplementary Video 1). **b**, 3D non-reciprocal motions (Supplementary Video 1). **c**, Reconfigurable motions (Supplementary Videos 2 and 3). **d**, Counter-rotating motions of two adjacent cilia (Supplementary Video 4). Scale bars, 8 μm (**a** (i)); 80 μm (**a** (ii)); 5 μm (**b** (i)); 40 μm (**b** (ii)); 5 μm (**c** (i)); 40 μm (**c** (ii)); 5 μm (**c** (iii)); 40 μm (**c** (iv)); 6 μm (**d** (i)); 50 μm (**d** (ii)).

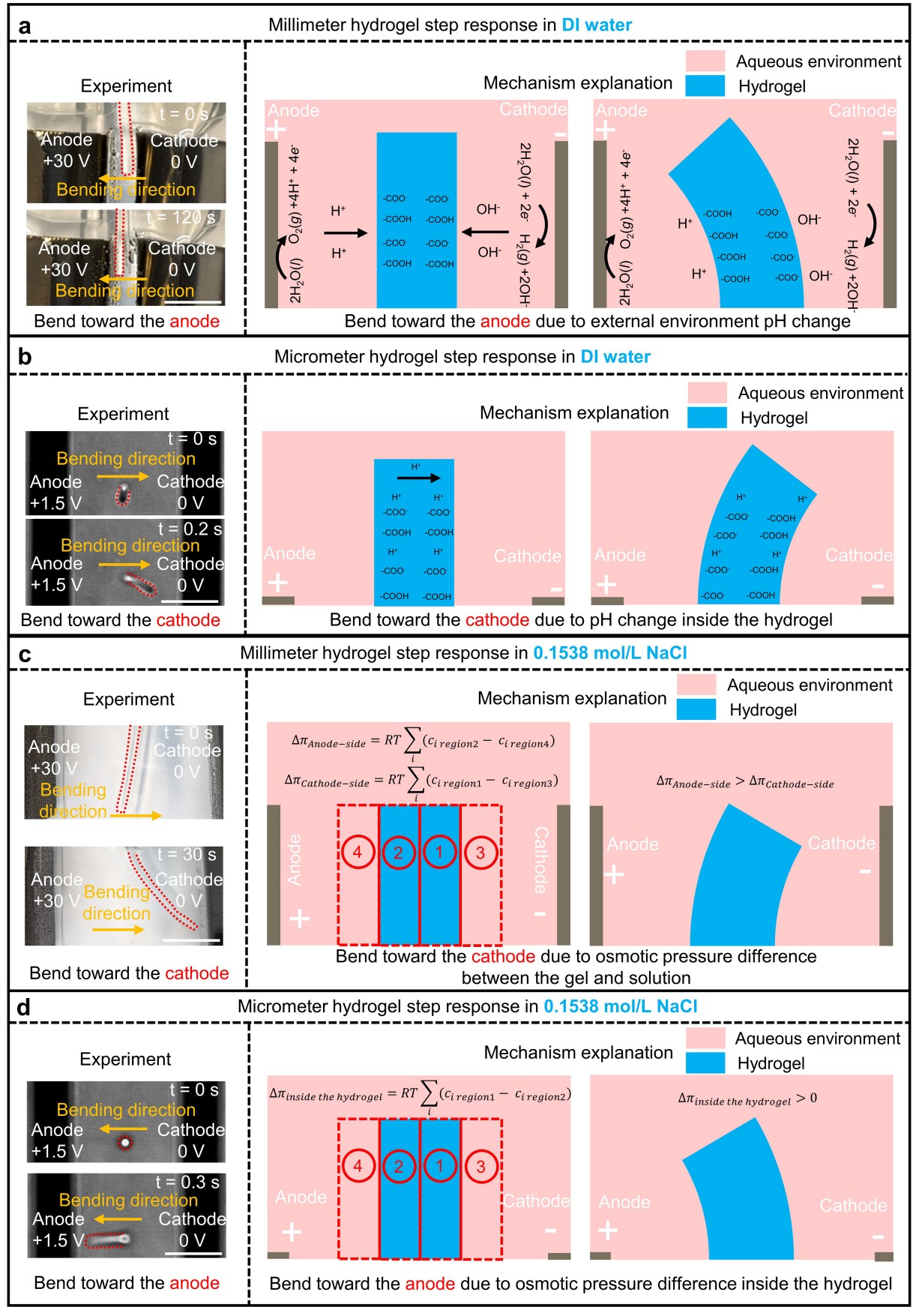

**Extended Data Fig. 5 | Working mechanism comparison between millimetre hydrogel and micrometre hydrogel. a**, The bending mechanism of millimetre hydrogel in DI water. **b**, The bending mechanism of micrometre hydrogel in DI water. **c**, The bending mechanism of millimetre hydrogel in 0.15380 mol l⁻¹ NaCl solution. **d**, The bending mechanism of micrometre hydrogel in 0.15380 mol l⁻¹ NaCl solution. Scale bars, 10 mm (**a**); 100 μm (**b**); 15 mm (**c**); 100 μm (**d**).

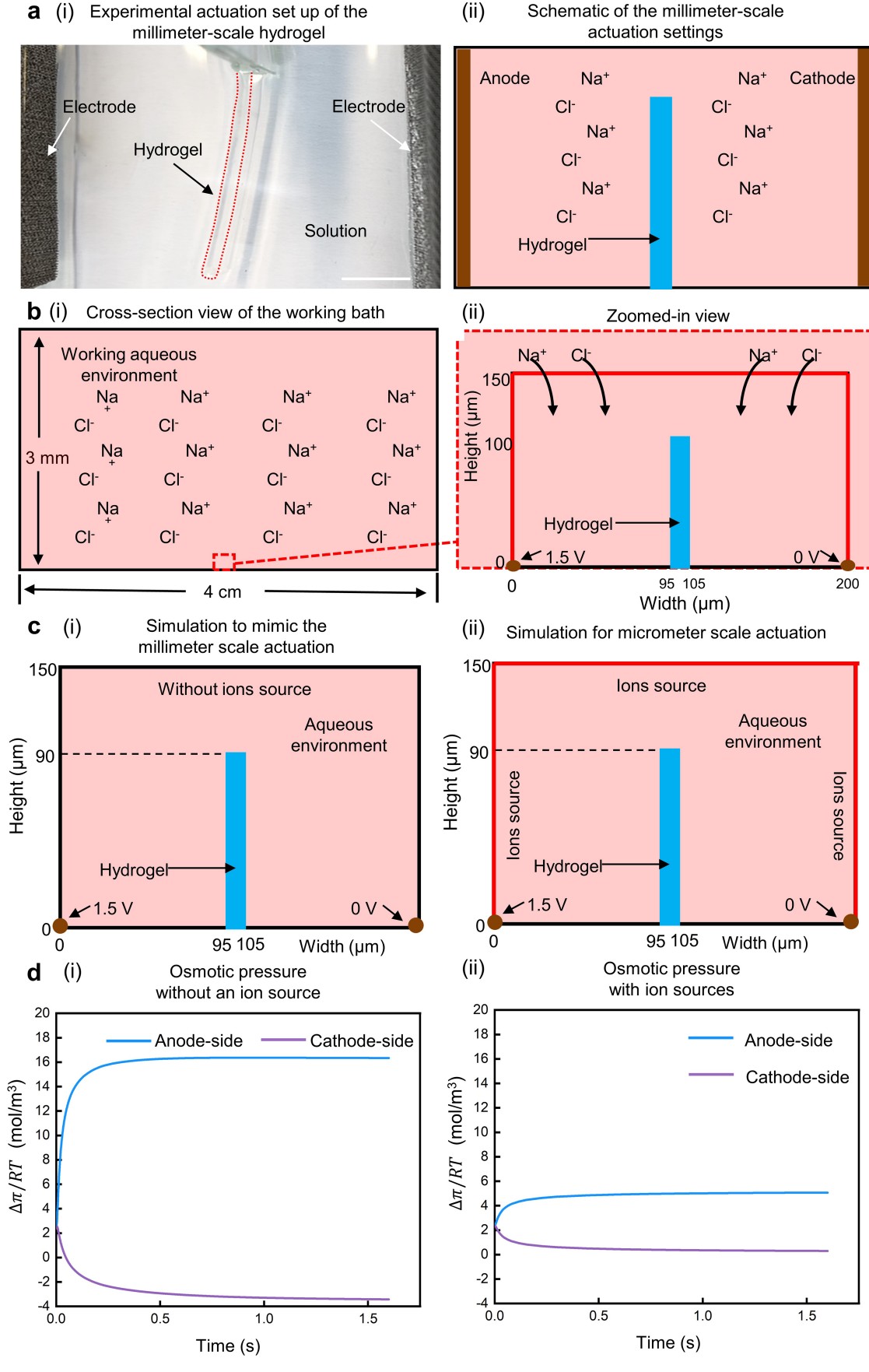

**Extended Data Fig. 6** | See next page for caption.

**Extended Data Fig. 6 | Osmotic pressure comparison between millimetre hydrogel and micrometre hydrogel. a**, (i) Photo of the millimetre-scale hydrogel actuation setting. (ii) Schematic illustrating the millimetre-scale hydrogel actuation setting. **b**, Schematics to show the working environments of the micrometre-scale hydrogel actuator. (i) Cross-sectional view of the gel microcilia working bath. The bath measures 4 cm (length) × 4 cm (width) × 3 mm (height) and is filled with solution, with NaCl solution used here as a typical example. The gel microcilia array and microelectrode array are integrated at the bottom of the bath. The hydrogel working region occupies a small portion of the bath, highlighted by the dashed red box. (ii) Enlarged view of the hydrogel working region. Two brown dots represent microelectrodes positioned at 0 μm and 200 μm, respectively. A voltage of 1.5 V is applied to the 0-μm electrode and 0 V to the 200-μm electrode. During actuation, mobile ions from the surrounding solution freely enter the working region, as indicated by black arrows. **c**, 2D simulation settings without and with ion sources. (i) Simulation settings to mimic the millimetre-scale actuation; there are no ion sources applied at the boundaries. (ii) Simulation settings for micrometre-scale actuation; ion sources are applied at the boundaries to mimic the external buffer ions. **d**, Osmotic pressure simulation in 0.01920 mol l$^{-1}$ NaCl. Here osmotic pressure refers to the interfacial osmotic-pressure difference that governs centimetre-scale and millimetre-scale hydrogel actuation (as defined in Extended Data Fig. 5c). (i) Osmotic pressure $\Delta\pi/RT$ data for the simulation without ion source. The anode-side osmotic pressure is much larger than the cathode-side pressure. This pressure difference bends the hydrogel towards the cathode and aligns with previous centimetre-scale hydrogel results[43]. (ii) Osmotic pressure data for the simulation with ion sources. The osmotic pressure difference between the anode and the cathode decreases substantially owing to the external buffer mobile ions.

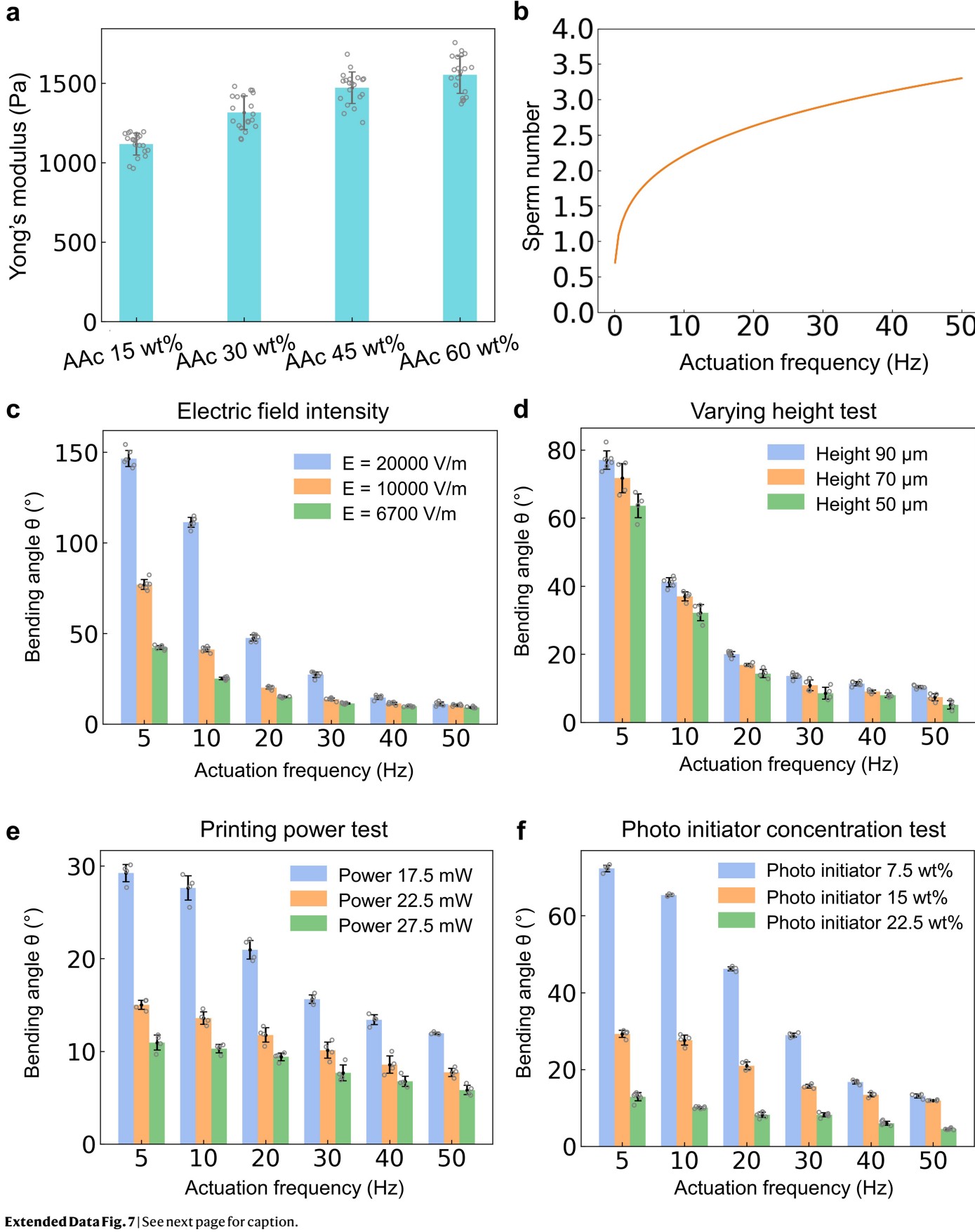

**Extended Data Fig. 7** | See next page for caption.

**Extended Data Fig. 7 | Further characterizations of TPP-printed hydrogels and hydrogel microactuators. a**, Young's modulus of hydrogels with varying AAc ratios. Data are presented as mean ± s.d. ($n$ = 20 tests). **b**, Sperm number of 30 wt% AAc hydrogel cilia at different actuation frequencies (diameter 10 μm, height 90 μm). **c**, Effect of electric-field intensity on the dynamic performance of the hydrogel microactuator. A 1.5-V potential is applied on electrodes with a distance between 75 and 230 μm to generate the corresponding electric-field intensity (30 wt% AAc, diameter 10 μm, height 90 μm, DI water). Data are presented as mean ± s.d. ($n$ = 6 samples). **d**, Effect of cilium height on dynamic performance. Cilia with diameters of 10 μm and heights of 90, 70 and 50 μm are tested (30 wt% AAc, DI water). Data are presented as mean ± s.d. ($n$ = 6 samples for 90 μm height, $n$ = 4 samples for 70 and 50 μm height). **e**, Effect of printing power on the dynamic performance of hydrogel actuators (45 wt% AAc, diameter 10 μm, height 90 μm, DI water). Data are presented as mean ± s.d. ($n$ = 4 samples). **f**, Effect of photoinitiator concentration on the dynamic performance of hydrogel actuators (15 wt% AAc, diameter 10 μm, height 90 μm, DI water). Data are presented as mean ± s.d. ($n$ = 4 samples for 7.5 wt% and 15 wt% photoinitiator, $n$ = 6 samples for 22.5 wt% photoinitiator).

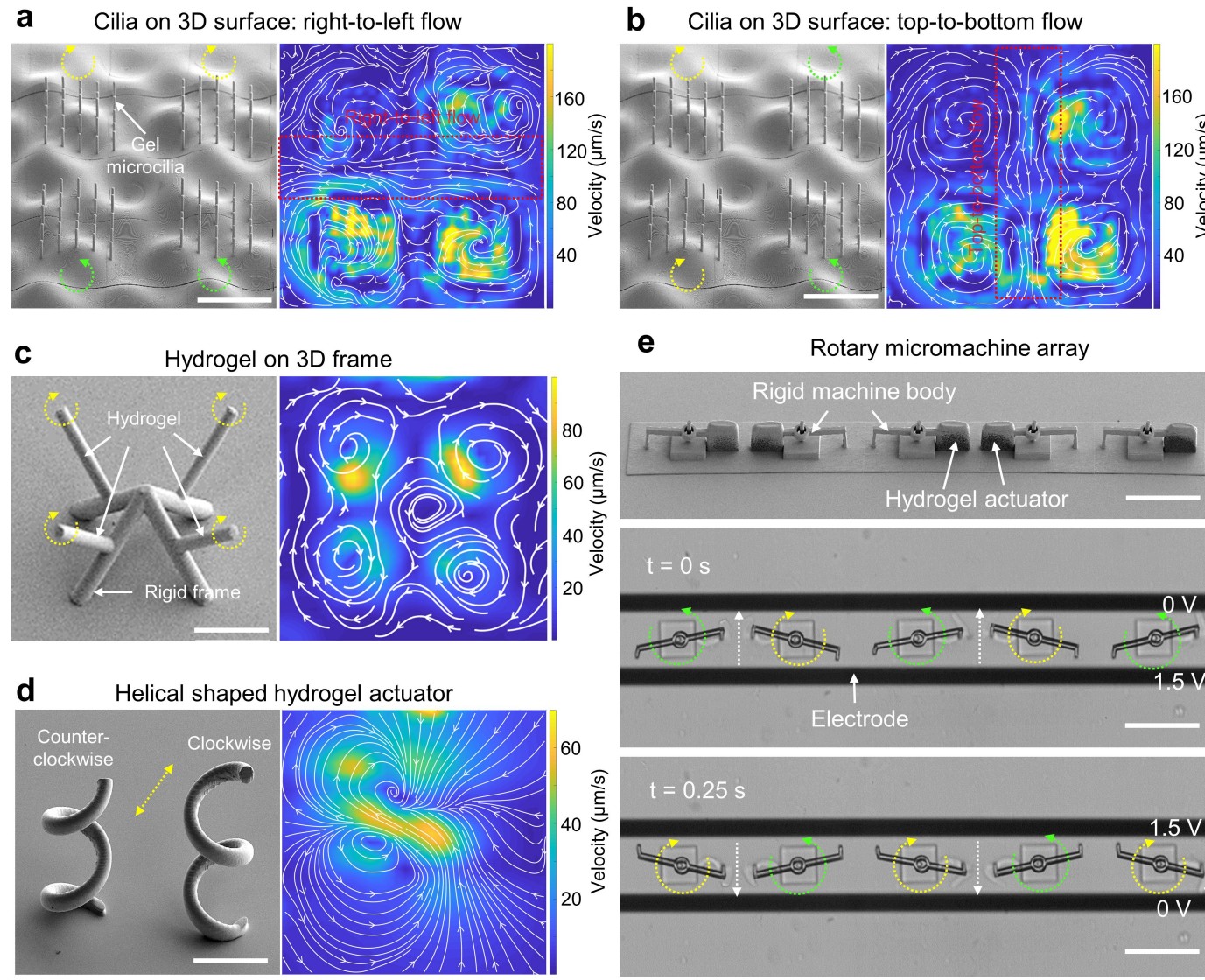

**a** Cilia on 3D surface: right-to-left flow

**b** Cilia on 3D surface: top-to-bottom flow

**c** Hydrogel on 3D frame

**d** Helical shaped hydrogel actuator

**e** Rotary micromachine array

**Extended Data Fig. 8 | Further configurations and application scenarios of the TPP-printed hydrogel microactuators. a,b,** Hydrogel cilia patterned on a 3D curved surface. **a,** Right-to-left flow on 3D surface. Left, SEM image with arrows indicating cilia rotation. Right, PIV result with the flow region highlighted by a red dashed box. **b,** Top-to-bottom flow on 3D surface. Left, SEM image with arrows indicating rotation directions. Right, PIV result with the flow region marked by a red dashed box. **c,** Four hydrogel cilia mounted on a 3D pyramid frame structure rotate synchronously to generate vortex flows. Left, SEM image showing the pyramid frame and hydrogel pillars, with arrows marking rotation directions. Right, PIV result of the flow field. **d,** Anticlockwise and clockwise helical hydrogel actuators. Left, SEM images with arrows marking motion directions. Right, corresponding PIV flow fields. **e,** Rotary micromachine array integrated with hydrogel microactuators. The first row is the SEM image of the array, with arrows marking actuator and machine positions. Following are video frames showing rotary motion, with arrows indicating the bending direction of hydrogel actuators and the joint rotation of the mechanical structure under actuation. Scale bars, 240 μm (**a**); 240 μm (**b**); 40 μm (**c**); 50 μm (**d**); 200 μm (**e**).

**Extended Data Table 1 | Comparison of the proposed gel microcilia in this work with the previously reported ciliary actuation works in the literature**

| Performance Index / Reported studies | Single cilium size | Maximum cilia numbers shown in the work | Reprogrammable coordinated motions | Motion frequency | Fluid manipulation capability | Durability test | Substrate |
|---|---|---|---|---|---|---|---|
| Pneumatic cilia[26] | cm | 6 | 2D reprogrammable motions | 0.25 Hz | Yes (0.5-1 mm/s) | N.A. | Rigid |
| Magnetic cilia[27-30] | mm to μm | ~$10^2$ | 2D and 3D motions | ~100 Hz | Yes (0.8 mm/s) | N.A. | Could be built on a stretchable substrate |
| Ultrasound-activated cilia[31,32] | μm | 10 | No | N.A. | Yes (up to 10 mm/s) | N.A. | Rigid |
| pH-sensitive cilia[33] | μm | ~$10^2$ | No | N.A. | N.A. | N.A. | Rigid |
| Light-responsive cilia[34,35] | μm | ~$10^2$ | 3D reprogrammable motions | ~0.1 Hz | N.A. | Tested 100 cycles[34] | Rigid |
| Electrostatically actuated cilia[36] | μm | ~$10^2$ | 2D reprogrammable motions | 200 Hz | Only in silicon oil (0.33-2 mm/s) | Tested 600,000 cycles | Rigid |
| Electro-chemically actuated cilia[37,38] | mm[37], and μm[38] | ~$10^2$ | 2D reprogrammable motions | ~10 Hz[37], and ~100 Hz[38] | Yes (3 mm/s[37], and maximum speed of 60 μm/s[38]) | Tested several thousand cycles in the works | Stretchable substrate[37], rigid substrate[38] |
| This work | μm | ~$10^6$ | 3D reprogrammable motions | 40 Hz | Yes (250 μm/s maximum speed) | Tested 330,000 cycles | Flexible polyimide substrate |

Data from refs. 26–38.

**Extended Data Table 2 | Material parameters of the Ogden model for hydrogels at different AAc concentrations**

| Parameters | 15 wt% AAc | 30 wt% AAc | 45 wt% AAc | 60 wt% AAc |
|---|---|---|---|---|
| $\mu$ (Pa) | 394 | 463 | 518 | 548 |
| $\alpha$ | 9.28 | 9.28 | 9.28 | 9.28 |
| $D$ (Pa$^{-1}$) | $8.60 \times 10^{-4}$ | $7.30 \times 10^{-4}$ | $6.53 \times 10^{-4}$ | $6.18 \times 10^{-4}$ |