## [Peer Review file · Nature]

3D-printed low-voltage-driven ciliary hydrogel microactuators

Corresponding Author: Professor Metin Sitti

Version 0:

Reviewer comments:

Referee #1

(Remarks to the Author)

This manuscript introduces ionic hydrogel-based microcilia actuated using an electrical field. Sizes of the cilia range from 2-10 micron in diameter and 18-90 micron in length, the smallest ones approaching the size of biological cilia. Two microcilia manufacturing approaches are used: direct 3D printing and micromolding. Arrays of cilia are integrated on a polyimide substrate that contains electrode patterns for applying time-dependent electric fields. Results show the microcilia deformation in various ionic solutions, how "reprogrammable" microcilia motions can be achieved using specific electrode activation protocols, and flow patterns generated by such motions.

The manuscript indeed contains some interesting technical innovations, and the technology may be useful for future microfluidic applications. Especially the capacity to control the motion of individual cilia is very nice, although the electrode layout will be complex for larger arrays. However, the novelty seems to be moderate. The concept of artificial cilia is not novel, the concept of "reprogrammability" of the motion of the arrays of cilia using electrical actuation was introduced in ref. 37, and the ionic hydrogel as an actuation material has been proposed and studied before. In addition, some of the results remain rather qualitative where a more quantitative analysis would have been possible. The manuscript may be suitable more for a more specialized journal. The following concrete points can be addressed.

- o In the explanation of the actuation mechanism, the electrostatic force seems to be ignored. One may expect that the ionic gel cilia have a non-zero zeta potential - the electrostatic force should at least be considered. Can the authors explain?
- o Most of the results, including those of the flow pattern generation, concern symmetrically rotating microcilia leading to vortical motions. To create net flow, asymmetric motions must be achieved. Can this be done with this technology? It seems relatively straightforward to do by applying different voltages to the electrodes surrounding the cilia, so that a time-dependent rotating electrical field magnitude is achieved causing a tilted conical motion. Can this be demonstrated?
- o In relation to this, the effect of the electrical field on the motion of the cilia is not quantified, e.g. in Fig. 2 – why not? This seems to be a control parameter that can be used in practice. Throughout the manuscript, only a few field magnitudes are used (but not compared).
- o The speed of the microcilia motion is only analyzed for a step response (Fig. 2); how about oscillations? Up to which frequency can the microcilia still follow the electrical field imposed and by which effect is this limited?
- o In the discussion of the combined effect of actuation frequency and microcilia size (Fig. 2Ciii), the influence of bending stiffness, inertia, and fluid drag is put forward. First, do inertial effects really play a role at all? A quick calculation of Reynolds number results in values smaller than 1 in all cases, which suggests that fluid flow inertial effects are absent. Second, the difference in bending between the two cilia sizes must be the result of the actuation force (ion motion) scaling differently with size than the counteracting forces (elasticity, viscosity) – it cannot be explained by considering only separate forces. How does that work?
- o Fig. 4 shows flow patterns generated by actuated arrays of microcilia, but the flow is not quantified. Since PIV measurements were carried out, the magnitude of the fluid flow velocities should be known. Why are these not reported? These are interesting for practical applications.
- o The abstract mentions that the presented microcilia enable "particle transportation". Is this really shown in the manuscript? Fig. 4, last column, shows particle tracking images, but these are particles that follow the fluid flow, aren't they? If so, the manuscript does not show direct particle transportation, but particles taken by fluid flow; any system that can generate fluid flow can cause such particle transportation and mentioning "particle transportation" in the abstract seems superfluous and

sets expectations that are not met.

o Fig. 4 shows interesting fluid patterns. What would be applications of such flows?

Referee #2

(Remarks to the Author)

I co-reviewed this manuscript with one of the reviewers who provided the listed reports.

Referee #3

(Remarks to the Author)

The manuscript discusses a ciliary platform using ion mitigation inside a hydrogel to instigate bending motions. Thorough insights are given on the mechanisms that drives this bending deformation, detailing the performance in DI and saline water. Carpets of cilia of different size and amount are fabricated that are either individually controlled or sectionally controlled. The capabilities of such cilia arrays have somewhat being demonstrated by means of imposing flow patterns.

To my feeling, the innovation of this manuscript lies in the fabrication process, creating cilia of micrometer dimensions, that span centimeter scaled surfaces. However the details of this process are being given in SI.

In contrast, the manuscript details on (i) differential swelling hydrogels, where basic mechanisms have already been detailed in literature (44-48); (ii) patches of cilia that show analogous modalities as displayed in (37). And although the results are well described, it is not where the real innovation lies.

Further, the manuscript lacks a clear demonstrator for such cilia.

Other critical questions:

- In the introduction, it is stated that current cilia are incompatible with ionic solutions (line 73). However the presented solution is also highly dependent on the surrounding fluid. In what sense is this solution capable of driving a wide variety of fluids (e.g. blood, saliva, urine, etc...). As such, the conclusion on line 240-241 seems to not well funded.
- In line 77 it is stated that soft substrates are beneficial, however throughout the manuscript, the softness of the substrate is not harnessed in applications.
- Is there an intuitive explanation for the fact that higher AAC concentrations produce smaller H⁺ ion concentration growth rate, besides simulation (line 140)?
- Regarding the reduced stroke at higher frequency, the dimensionless sperm number is typically used to describe this phenomena.
- In SI movie 2, it seems that the movie in DI water is cut short, as it seems to bend backwards towards the anode. Can you provide a longer movie, and explain if this is happening?
- The manuscript has grammatical mistakes (e.g. line 113 misses a verb), and should be revised in this sense.
- A formal conclusion is lacking.

Referee #4

(Remarks to the Author)

The authors present a scalable, high-precision, and electrically controllable soft-body ciliary array system for microscale fluid manipulation. The work is particularly impressive in demonstrating programmable 3D motion with fast response times and independent actuation of individual cilia. The concept is well-articulated, and the manuscript is written clearly with a broad audience in mind. I find the study to be both innovative and significant. I recommend major revision to strengthen the manuscript before publication. Please see my comments below:

1. To fully demonstrate the programmability and robustness of the ciliary system, additional quantitative characterization is needed. While the effects of NaCl concentration, cilia diameter, and solution properties are examined, the influence of electric field intensity and cilia height on actuation behavior should studied.
2. The authors should elaborate on potential applications of the system. Providing at least one concrete example (e.g., microfluidic mixing, targeted cargo transport, or biomedical sensing) would highlight the broader impact of the technology.
3. Can the authors provide data (experimental or simulated) on the mechanical strength of the hydrogel cilia and estimate the magnitude of the electric field-induced forces? This would clarify the relationship between applied field and resulting displacement.
4. Given the high electric field used, please address whether water ionization or electrolysis occurs under experimental conditions. This could have implications for long-term stability or application in biological systems.
5. The definition of bending angle should be clarified. Please specify how bending is quantified in this work.
6. Reduced bending amplitudes may also be influenced by the Young's modulus of the hydrogel. Please discuss whether variations in polymer crosslinking or photo-initiator concentration affect mechanical stiffness, and how that might influence actuation amplitude.
7. The mechanism by which the hydrogel responds to the electric field requires more explanation. A deeper discussion—supported by references and your own characterizations—would benefit readers unfamiliar with electroactive gels.
8. A mechanical model or analysis of the deformation behavior would significantly enhance the study. Even a simplified analysis could provide valuable insights into the bending mechanism and design parameters.
9. Please clarify the polarity and direction of the applied field in Fig. 1C, and revise the region labeling (Region 1 vs. Region

2) to avoid confusion. A clearer schematic, as seen in Supplementary Fig. S13, would be helpful here.

10. If AAC content plays a key role, why do the bending angles in Fig. 2C show minimal variation across groups at higher frequencies (e.g., 20–50 Hz)? Statistical analysis is needed to confirm trends. Additionally, can the authors provide Young's modulus measurements for cilia of different diameters?

12. A schematic of the electrode configuration used in Fig. 3 would help clarify the observed 180° phase shift and its relation to cilia motion.

13. Figures 2biii–vi show that tip displacement appears insensitive to NaCl concentration, while bending angle is affected. Meanwhile, Fig. 2Civ shows bending angle varies with frequency. Could the authors clarify the relationship between tip displacement and bending angle? Are these quantities independently controlled or correlated?

14. In Figure 4, please include a scale bar to indicate the magnitude of the velocity field.

15. The table is incomplete; please include the corresponding fluid flow velocity values.

16. Since the article focuses on ciliary actuation, the authors may consider recent developments in ultrasound-driven artificial cilia (<https://doi.org/10.1073/pnas.2418938122>). A brief comparison could help contextualize the present work in relation to other actuation modalities.

Version 1:

Reviewer comments:

Referee #1

(Remarks to the Author)

The authors have addressed all my previous comments in a substantial, detailed, and satisfactory manner. In particular, the novelty of the work has been clarified much; indeed, the manuscript now convincingly describes that it is the combination of the novel actuation mechanism that enables fast actuation at low voltages, effective in diverse fluidic environments, with the 2PP fabrication method that enables the creation of hydrogel microactuators that operate according to this mechanism. The new title reflects this well. The additional demonstrations of new applications support the relevance of the reported innovations.

Also, all of my concrete remarks and concerns have been addressed well, and they have led to revisions of the manuscript which clarify these points well.

Taken together, I believe that in the current form the manuscript is acceptable for publication.

Referee #2

(Remarks to the Author)

I co-reviewed this manuscript with one of the reviewers who provided the listed reports.

Referee #3

(Remarks to the Author)

I am satisfied with the way the authors addressed my remarks and have no further comments.

Referee #4

(Remarks to the Author)

The authors have provided thorough and convincing responses to the reviewers' comments. I am satisfied that the concerns have been fully resolved and recommend the manuscript for publication in Nature.

Response Letter

We sincerely thank the Editor and Reviewers for their thoughtful and constructive comments, which improved our manuscript significantly. A detailed point-by-point response is provided below. The comments we received are colored with blue, our responses are followed in black font, and the changes to the original manuscript and supplementary information are highlighted in yellow in the revised manuscript.

Comments from the Editor

Your manuscript, "Electrically Actuated Ultra-soft Hydrogel Microcilia for Dynamic Fluid Manipulation", has now been seen by 4 referees, whose comments are attached below. While they find your work of potential interest, as indeed do we, they have raised substantial concerns that first need to be addressed before we can consider the paper further for possible publication in Nature.

In view of the additional experimental data and theoretical analysis that are required to address these concerns, we appreciate that the necessary revisions will probably take some time. But let me assure you that we will nevertheless be happy to look at a revised manuscript (unless, of course, something similar has by then been accepted at Nature or appeared elsewhere). We hope to receive your revised paper within four months. If you cannot complete the revisions within this time frame, please let us know when you would anticipate being able to submit a revised manuscript.

In particular, we would expect such revised manuscript can fully demonstrate the applicational advances.

Response:

We sincerely thank the Editor for the summary and the Reviewers for their thoughtful and constructive comments. In revising the manuscript, we engaged in extensive reflection, discussion, and additional experimental investigation. These efforts led to new insights regarding both the actuation mechanism questioned by **Reviewer 1** ("the ionic hydrogel as an actuation material has been proposed and studied before") and the fabrication contributions highlighted by **Reviewer 3** ("the innovation of this manuscript lies in the fabrication process"). Before addressing specific points in detail, we would like to clearly outline the main contributions and novelties of this work.

1. Biologically analogous ciliary motion

Reviewers 1 and 3 noted that programmability of artificial cilia arrays had been investigated previously (1-7), with earlier reports describing reprogrammable motion under electrical actuation (e.g., Ref. 37). As summarized in Fig. R1, while previous artificial cilia studies have made some progresses, they did not reproduce critical features of biological ciliary motion (Fig. R2 (Fig. S5 in the revised supplementary), left column, adapted from (8)). In contrast, our hydrogel microcilia closely resemble biological cilia (Fig. R2 (Fig. S5 in the revised supplementary), right column).

This advance was in fact acknowledged by **Reviewer 1** (“the capacity to control the motion of individual cilia is very nice”) and **Reviewer 4** (“particularly impressive in demonstrating programmable 3D motion with fast response times and independent actuation”).

2. Novel actuation mechanism at the microscale, enabled by two-photon lithography

The key to this performance lies in a fundamentally distinct actuation mechanism. Our microscale hydrogels exhibit bending behavior opposite to that of their millimeter-scale counterparts (Fig. R3Di, left; Fig. R3Dii; Movie S9) (9-12). This constitutes the direct evidence that hydrogels at different scales operate through fundamentally different mechanisms.

Millimeter scale: **Bending arises from ion migration across the gel–solution interface.**

- In DI water: driven by bath-induced pH gradients caused by electrolysis.
- In ionic solutions: driven by interfacial osmotic pressure from ion partitioning.

Micrometer scale: **Bending arises from cation migration within the hydrogel network along electric double layers (EDLs) around the negatively charged ($-\text{COO}^-$) networks.**

- In DI water: mobile H^+ ions (from dissociation of $-\text{COOH}$) migrate inside the hydrogel, establishing internal pH gradients that contract the network and leads to deformation.
- In ionic solutions: mobile Na^+ ions migrate inside the hydrogel, inducing electroosmotic flow; the associated water transport swells the network and leads to deformation.

Further details are provided in **Comment 2, Reviewer 1**, and Fig. R3D.

The core of the micrometer-scale hydrogel actuation mechanism lies in the enhanced electric double layer (EDL) by TPP fabrication: We found that conventional molding or flood-exposure techniques typically produce ionic hydrogels with micrometer-scale pores (SEM results in Fig. R3Bi) (13). By contrast, our use of two-photon polymerization with carefully engineered polymerization pathways (Fig. S8 in the revised supplementary) and low printing power (~ 15 mW) achieves nanometer-scale porosity (TEM results in Fig. R3Bii) while maintaining well-defined morphological features. This nanoscale porosification increases the effective surface area and expands the capacity of the electric double layer, thereby enhancing ion transport and electroosmotic flow in ionic solutions (14-16).

As a consequence of the newly established mechanism and engineering approach shown above, the response time shortened from 30 s (Fig. R3Di, left) to 0.3 s (Fig. R3Dii), while bending amplitude increased by 4–5 times compared with miniature actuators fabricated by molding (Fig. R3Di, right).

3. Applicational advances enabled by this mechanism

Building upon these mechanistic and fabrication innovations, we demonstrate several new applications that go beyond prior hydrogel actuation approaches:

- Dynamic flow control on complex 3D curved surfaces (Fig. R4A), enabling operation in realistic, non-planar environments (Movies S17-18).
- A biomimetic starfish-larva platform (Fig. R4B) (17), which highlights the system's utility for biomechanical and microrobots propulsion studies (Movie S21).
- Diverse actuator geometries integrated with micromachines (Fig. R4C), showcasing the versatility and engineering potential of the approach (Movies S22-23).
- Being functional across multiple physiological fluids, such as human saliva, human serum, and mouse plasma (Movie S3).

Summary

In this work, we establish a new low-voltage actuation mechanism at the microscale, enabled by two-photon polymerization–assisted fabrication. This strategy allows hydrogel cilia to replicate key features of biological motion with high fidelity. It also delivers a generalizable technique for directly printing miniature hydrogel actuators with superior speed, strength, and versatility beyond ciliary systems (e.g., as shown on the new flapping wing micromachine example in Fig. 3 of revised manuscript). With this consideration, we also adapted our title to “3D-printed, Low Voltage-Driven Ciliary Hydrogel Microactuators.”

Redacted

Fig. R2 (Fig. S5 in the revised supplementary). Comparison between biological cilia and artificial hydrogel cilia. The left column shows mouse ventral nodal cilia (with permission from

the publisher) (8), and the right column shows the hydrogel cilia developed in this work. **(A)** Cilia arrays (Movie S1). **(B)** 3D non-reciprocal motions (Movie S1). **(C)** Reconfigurable motions (Movies S2-3). **(D)** Counter-rotating motions of two adjacent cilia (Movie S4). Scale bars: **(A)** **(i)** 8 μm , **(ii)** 80 μm , **(B)** **(i)** 5 μm , **(ii)** 40 μm , **(C)** **(i)** **(iii)** 5 μm , **(ii)** **(iv)** 40 μm . **(D)** **(i)** 6 μm , **(ii)** 50 μm .

A i Conventional flood-exposure fabrication

ii TPP printing fabrication

B i SEM image of the flood exposure fabricated hydrogel

ii Image of the TPP printed hydrogel

C i Flow in the micrometer hydrogel channel under electric field

ii Flow in the nanometer hydrogel channel under electric field

D i Dynamic motion of the flood exposure fabricated hydrogel

ii Dynamic motion of a TPP printed micrometer scale hydrogel

Fig. R3 (Fig. S1 in the revised supplementary). Comparison between hydrogels fabricated by conventional flood exposure and two-photon polymerization (TPP). (A) (i) Schematic of the conventional flood-exposure fabrication process. (ii) Schematic of the TPP fabrication process, which allows tuning of printing by adjusting parameters such as hatching and slicing. (B) (i) SEM image of a hydrogel fabricated by flood exposure, showing pore sizes in the range of 10–30 μm . (ii) SEM and TEM images of the TPP-printed hydrogel. At this magnification, the SEM image shows a nearly featureless surface without discernible pores. The TEM image reveals pore sizes ranging from 20 to 80 nm. (C) Schematics illustrating the influence of pore size on electroosmotic flow. (i) Flow in a micrometer-scale ionic hydrogel channel. (ii) Enhanced flow in a nanometer-scale ionic hydrogel channel. (D) Dynamic performance of hydrogels fabricated by different methods. (i) Millimeter-scale (left) and micrometer-scale (right) hydrogels fabricated by flood exposure. (ii) Micrometer-scale hydrogel fabricated by TPP printing.

A Cilia patterned on a hill-shaped surface

B Biomimetic artificial starfish larva

C Other hydrogel microactuator variant

Fig. R4. Application scenarios of the hydrogel microactuator. (A) Cilia patterned on a hill-shaped surface. The surface height difference is 100 μm . (i) Coordinated motions of the cilia could generate a right-to-left flow on the 3D surface. Left: SEM image with arrows indicating cilia rotation; Right: PIV result with the flow region highlighted by a red dashed box. (ii) Top-to-bottom flow. Left: SEM image with arrows indicating rotation directions; Right: PIV result with the flow region marked by a red dashed box. (B) Biomimetic artificial starfish larva. (i) Photo of artificial starfish larva with patterned electrodes prior to cilia integration. (ii) SEM image of larvae with distributed cilia; arrows indicate rotation directions. (iii) PIV result of the flow field generated by cilia motion. (iv) Schematic showing the flow generated by cilia motion in biological starfish larva (17). (C) Other hydrogel microactuator variant. (i) Four cilia mounted on a 3D pyramid frame structure rotate synchronously to generate vortex flows. Left: SEM image showing the pyramid frame and hydrogel pillars with arrows marking rotation directions; Right: PIV result of the flow field. (ii) Counterclockwise and clockwise helical hydrogel actuators. Left: SEM images with arrows marking motion directions; Right: corresponding PIV flow fields. (iii) Rotary micromachine array with integrated hydrogel actuators. The first row is the SEM image of the array, with arrows marking actuator and machine positions. Following are video frames showing rotary motion, with arrows indicating the bending direction of hydrogel actuators and the joint rotation of the mechanical structure under actuation. (iv) Flapping mechanism integrated with hydrogel actuators. The first row is the SEM image with arrows marking actuator and machine positions. Following are video frames showing flapping motion, with arrows indicating the bending direction of hydrogel actuators and the corresponding joint motion of the mechanical structure. Scale bars: (A) (i–ii) 240 μm , (B) (i) 400 μm , (ii) 360 μm , (C) (i) 40 μm , (ii) 50 μm (iii) 200 μm , (iv) 100 μm .

Reviewer #1

Overall Comment:

This manuscript introduces ionic hydrogel-based microcilia actuated using an electrical field. Sizes of the cilia range from 2-10 micron in diameter and 18-90 micron in length, the smallest ones approaching the size of biological cilia. Two microcilia manufacturing approaches are used: direct 3D printing and micromolding. Arrays of cilia are integrated on a polyimide substrate that contains electrode patterns for applying time-dependent electric fields. Results show the microcilia deformation in various ionic solutions, how “reprogrammable” microcilia motions can be achieved using specific electrode activation protocols, and flow patterns generated by such motions.

The manuscript indeed contains some interesting technical innovations, and the technology may be useful for future microfluidic applications. Especially the capacity to control the motion of individual cilia is very nice, although the electrode layout will be complex for larger arrays.

Response:

After careful consideration of the concern that our contributions may appear technical, we would like to clarify that **the novelty of this work lies in the actuation mechanism itself, which is fundamentally distinct from previously reported hydrogel actuation approaches (9-12)**. Moreover, our use of the two-photon polymerization (TPP)-based 3D printing approach to realize this mechanism is also novel for hydrogel microactuator fabrication. Therefore, we appreciate the reviewer’s comment, as it provided us the opportunity to highlight this point more clearly in the revised manuscript. Please kindly check the summary in our **Response to the Editor** at the beginning.

Regarding the observation that “**the electrode layout will be complex for larger arrays,**” we agree that scaling increases routing complexity. Meanwhile, established microelectronics fabrication technologies can reliably produce highly intricate, multilayer microscale electrode architectures (18, 19). Our focus here is on the microscale actuation mechanism itself, while scalable electrode architectures are compatible with standard processes and can be pursued in future system-level implementations.

Major Comment 1:

However, the novelty seems to be moderate. The concept of artificial cilia is not novel, the concept of “reprogrammability” of the motion of the arrays of cilia using electrical actuation was introduced in ref. 37.

Response:

We have clarified that our primary contribution lies in the actuation mechanism itself, and the use of the TPP approach to realize this mechanism. Please refer to our **last response** above and our **Response to the Editor** at the beginning.

While ref. 37 (I) discussed electrically reprogrammable motions in artificial cilia arrays, it is important to note that “reprogrammability” originates as an intrinsic property of biological cilia (Fig. R2 (Fig. S5 in the revised supplementary), the left column) (8). And the “reprogrammability” feature had been explored in earlier electrically actuated artificial systems as well (7). In this continuum, ref. 37 (I) contributed valuable demonstrations with 2D beating beam (Fig. R1F), and **our study advances the field even further to closely mimic the real cilia in terms of 3D reconfigurable motion (Fig. R2 (Fig. S5 in the revised supplementary))**.

In addition to biologically analogous ciliary motion and reprogrammability, we also have three more advantages.

1. **Durability (life cycle time) comparison.** The operational lifetime of artificial cilia is also a challenge for ref. 37 (I). Because SEA actuators rely on continuous redox reactions on the surface of the thin-film actuator, stable performance is limited to only a few thousand cycles (Fig. R5A). At high frequencies, this translates into only a few minutes of reliable operation. By contrast, our hydrogel cilia are actuated purely by ion migration within the hydrogel network (Fig. R5B), without involving electrochemical reactions (please refer to our Response to Major Comment 2 for details on the mechanism). We continuously tested our hydrogel cilia for 330,000 cycles (~5 hours at 20 Hz), and they maintained stable performance (Fig. R5B). It should be noted that after ~80,000 cycles, the bending amplitude decreased to ~70% of the initial value, but thereafter the performance stabilized and did not degrade further (Movie S11).
2. **Working environment comparison.** SEA cilia in ref. 37 (I) require ionic environments to support necessary redox reactions and were demonstrated only in PBS solution. By contrast, our hydrogel cilia, due to the presence of dissociable H⁺ ions in the hydrogel network itself, can function in both deionized water—relevant for applications requiring low ionic strength, such as Raman measurements or particle transport for surface charge analysis—and in a wide range of ionic solutions, suitable for scenarios such as cell transport and sorting. In this revision, we further evaluated their performance in multiple biofluids, including human saliva, human serum, and mouse plasma (Fig. R6). In all cases, the hydrogel cilia maintained stable actuation, demonstrating broader versatility and potential applicability across diverse environments.
3. **Can be 3D-printed on 3D surfaces.** We integrated the polyimide substrate, microelectrodes, and hydrogel actuators onto a three-dimensional curved surface with a height variation of 100 μm (Fig. R7, Movies S17-18). The hydrogel actuators exhibited stable motion and effective fluid manipulation on this curved geometry. As shown in Fig. R7, coordinated hydrogel motion enabled controllable flows along both right-to-left (Fig. R7A) and top-to-bottom (Fig. R7B) directions on the 3D surface. The three-dimensional surface more closely resembles the non-flat morphology of physiological substrates and also provides engineering opportunities beyond those offered by previously reported approaches (Fig. R1).

A

[PANELS REDACTED]

Fig. R5. Comparison between the SEA cilia (R1B3) and hydrogel cilia.

Fig. R6. Hydrogel cilia working in different solutions

Fig. R7 (Fig. S25 in the revised supplementary). Cilia patterned on a 3D curved surface. (A) Right-to-left flow on 3D surface. Left: SEM image with arrows indicating cilia rotation; Right: PIV result with the flow region highlighted by a red dashed box. (B) Top-to-bottom flow on 3D surface. Left: SEM image with arrows indicating rotation directions; Right: PIV result with the flow region marked by a red dashed box.

Major Comment 2:

And the ionic hydrogel as an actuation material has been proposed and studied before. In addition, some of the results remain rather qualitative where a more quantitative analysis would have been possible. The manuscript may be suitable more for a more specialized journal. The following concrete points can be addressed.

Response:

The mechanism reported previously is different from what is reported here (Fig. R3D).

Previously, ionic hydrogels indeed have been investigated as actuator materials, but primarily at the millimeter–centimeter scale (9-12). The present work advances this field to the microscale and contribute in two key aspects:

1. We provide the systematic study of these TPP printed ionic-hydrogel actuators, and find that the governing mechanisms differ fundamentally from those at millimeter scales (see our Response to the Editor at the beginning, Table R1).

2. We establish a quantitative framework that integrates controlled experiments with a fully coupled electro-chemo-mechanical model to explain microscale ionic-hydrogel bending. In this model, the influence of densely distributed -COO^- groups on ion migration within the hydrogel is included, capturing the influence of the nanometer-scale hydrogel channels.

Full details of the mechanism, modeling, and simulation are provided in the **supplementary “Coupled electro-chemo-mechanical model”** section, which is also detailed below:

1. Comparison of actuation mechanism between micro and millimeter scale hydrogel (Fig. R8-R11)

Various bending of ionic-hydrogel actuators under electrical stimulation has been characterized at larger scales (9-12) and our experiments show that the microscale regime exhibits intrinsically different dynamics. For example, in DI water, a millimeter-scale actuator bends toward the **anode** in ~ 120 s (Fig. R8, Movie S9), whereas a micrometer-scale actuator bends toward the **cathode** in ~ 0.2 s (Fig. R9, Movie S9). In 0.1538 mol/L NaCl, the millimeter-scale actuator bends toward the **cathode** in ~ 30 s (Fig. R10, Movie S9), while the micrometer-scale actuator bends toward the **anode** in ~ 0.4 s (Fig. R11, Movie S9). These differences are summarized in Table R1. In summary, micrometer-scale hydrogels bend in the opposite direction to their millimeter-scale counterparts and respond over two orders of magnitude faster—differences not explained by existing theories (9-12). The detailed mechanism difference is analyzed below:

1.1 Mechanism difference in DI water

Millimeter-scale mechanism in DI water is shown Fig. R8. Under the applied voltage, electrolysis produces H^+ near the anode and OH^- near the cathode. Because the AAc-co-AAm network is pH-sensitive, acidic conditions convert -COO^- to -COOH (reducing repulsion force and causing contraction), whereas alkaline conditions convert -COOH to -COO^- (increasing repulsion and causing swelling). The bath-induced pH gradient therefore shrinks the anode-side region and swells the cathode-side region, resulting in bending toward the anode.

Micrometer-scale mechanism in DI water is shown Fig. R9. In contrast, the micrometer-scale hydrogel bends toward the cathode. Here, fixed -COOH groups partially dissociate into -COO^- and mobile H^+ . To maintain electroneutrality, the dissociated H^+ ions largely remain confined within the hydrogel. Under an external field, these H^+ ions migrate and accumulate on the cathode-facing side, locally lowering pH and inducing network contraction, which bends the hydrogel toward the cathode.

1.2 Mechanism difference in 0.1538 mol/L NaCl

Millimeter-scale mechanism in 0.1538 mol/L NaCl is shown in Fig. R10. In a saline environment, the external field drives the migration of all mobile ions in the bath. The hydrogel's fixed negative charges influence ion partitioning at the gel–solution interface (9-11), producing non-uniform ion concentrations across four regions (gel anode side: region 2; gel cathode side: region 1; solution anode side: region 4; solution cathode side: region 3). According to Flory's

theory (20), the local osmotic pressure difference is $\Delta\pi = RT \sum_i (c_{ig} - c_{is})$, where $\Delta\pi$ is the pressure difference, c_{ig} is the ion concentration in the hydrogel, c_{is} is ion concentration in solution, R is the gas constant, and T is temperature. Under steady-state condition, the osmotic pressure difference on the anode side $\Delta\pi_{Anode-side} = RT \sum_i (c_{i\ region2} - c_{i\ region4})$ exceeds that on the cathode side $\Delta\pi_{Cathode-side} = RT \sum_i (c_{i\ region1} - c_{i\ region3})$. This imbalance causes the anode side to swell more, bending the hydrogel toward the cathode.

Micrometer-scale mechanism in 0.1538 mol/L NaCl is shown in Fig. R11. The dominant factor is the ion concentration gradient inside the hydrogel rather than at the interface. The relevant osmotic pressure is $\Delta\pi_{inside\ the\ hydrogel} = RT \sum_i (c_{i\ region1} - c_{i\ region2})$, which remains positive, making the cathode side swell more and driving bending toward the anode.

2 Quantitative framework for micrometer hydrogel actuation mechanism (Fig. R12)

In this revision, we developed a fully coupled electro-chemo-mechanical model and performed corresponding simulations. The model simultaneously resolves (i) ionic concentration distribution under an external electric field and the fixed charges of the hydrogel network, (ii) the resulting osmotic body forces generated by non-uniform ion distributions, and (iii) the gel deformation driven by these body forces. At this stage, the model focuses on explaining the deformation of hydrogels in ionic solutions under external electric fields, and does not include fluid–structure interactions between the gel and the surrounding liquid.

As shown in Fig. R12, the simulations reproduce our experimental observations across representative media: in DI water, H^+ released by the gel dominate the bending, and the microhydrogel bends toward the cathode under external electric field (Fig. R12A vi-ix); in physiological saline (0.1538 mol/L NaCl), abundant Na^+ dominates, and the gel bends toward the anode (Fig. R12B iv). The agreement between simulations and experiments in both media supports the proposed microscale deformation mechanism.

The model further predicts—and the experiments confirm—a competition between H^+ and Na^+ that leads to a transient reversal of bending direction at intermediate ionic strength (Fig. R12B iii). Specifically, in 0.00769 mol/L NaCl, the hydrogel first bends toward the cathode and subsequently switches toward the anode under an electric field. This behavior is consistent with the higher mobility of H^+ relative to Na^+ : H^+ ions arrive first to induce local network contraction, followed by the slower Na^+ influx that offsets H^+ -induced contraction and reverses the bending direction. Simulations capture this sequence and match the experimental time course (Fig. R12B i), further validating the proposed mechanism.

Hydrogel actuator types		Reported large scale (cm-mm) hydrogel (9-12)	This work (μm)
Working environment			
DI water	Bending direction	Bend toward the anode	Bend toward the cathode
	Mechanism	External environment pH change	pH change inside the hydrogel
	Actuation voltage	Tens of volt (strong electrolysis reaction)	1.5 V (no electrolysis reaction)
	Time to reach equilibrium	120 s	0.2 s
Ionic solutions (0.1538 mol/L NaCl)	Bending direction	Bend toward the cathode	Bend toward the anode
	Mechanism	Osmotic pressure difference between the gel and solution	Osmotic pressure difference inside the hydrogel
	Actuation voltage	Tens of volt (strong electrolysis reaction)	1.5 V (no electrolysis reaction)
	Time to reach equilibrium	30 s	0.4 s

Table R1. Comparison between previous large scale ionic hydrogel actuator and micro hydrogel actuator of this work.

Fig. R8 (Fig. S9 in the revised supplementary). Millimeter hydrogel step response in DI water. Scale bar: 10 mm.

Fig. R9 (Fig. S10 in the revised supplementary). Micrometer hydrogel step response in DI water. Scale bar: 100 μm .

Fig. R10 (Fig. S11 in the revised supplementary). Millimeter hydrogel step response in 0.1538 mol/L NaCl. Scale bar: 15 mm.

Fig. R11 (Fig. S12 in the revised supplementary). Micrometer hydrogel step response in 0.1538 mol/L NaCl. Scale bar: 100 μm .

Fig. R12 (Fig. 2A-B in the revised manuscript). Experiments and fully coupled simulations. (A) Step response of hydrogel microcilia in DI water. (i) Time sequence of a 30 wt% AAc gel microcilium bending in DI water, with the left electrode as the anode and the right as the cathode. The cilia are outlined with dashed lines to aid visualization. Under the applied signal, the cilium bent toward the cathode. (ii–v) Experimental tip displacement of gel microcilia with varying AAc concentrations, showing that higher AAc content reduces tip displacement under the same step input. (vi–ix) Simulated bending of gel microcilia with different AAc concentrations. The simulations reproduce the experimental trend, with bending amplitude decreasing as AAc content

increases. **(B)** Step response of hydrogel microcilia in NaCl solutions of different concentrations. **(i)** In 0.00769 mol/L NaCl, the gel microcilium first bent toward the cathode and then gradually reversed to bend toward the anode. **(ii)** In 0.1538 mol/L NaCl, the gel microcilium bent only toward the anode. **(iii–vi)** Simulated bending in 0.00769 mol/L and 0.1538 mol/L NaCl solutions. Consistent with the experiments, the simulation in 0.00769 mol/L NaCl shows initial bending toward the cathode followed by reversal toward the anode, while in 0.1538 mol/L NaCl the bending occurs only toward the anode. Scale bars: **(A)** **(i)** 100 μm , **(B)** **(i)** **(ii)** 100 μm .

Specific Comment 1:

1. In the explanation of the actuation mechanism, the electrostatic force seems to be ignored. One may expect that the ionic gel cilia have a non-zero zeta potential - the electrostatic force should at least be considered. Can the authors explain?

Response:

We thank the reviewer for raising this point. As noted, the hydrogel used in this work carries a negative charge due to the fixed $-\text{COO}^-$ groups in its polymer network. To address this, we measured the zeta potential of the 30 wt% acrylic acid hydrogel under different solution conditions (Fig. R13 (Fig. S32 in the revised supplementary)). The hydrogel exhibited a zeta potential of -19.1 mV in DI water, which increased with NaCl concentration, reaching -3.7 mV at 0.1538 mol/L NaCl.

Our experimental results, however, indicate that electrostatic force does not govern hydrogel microcilia bending. **In DI water, where electrostatic effects should be strongest, the negatively charged hydrogel microcilia bent toward the cathode (Fig. R3A).** If electrostatic attraction were dominant, bending toward the anode would be expected. This opposite behavior suggests that electrostatic interactions play a negligible role in the observed deformation.

We further compared the magnitudes of the two forces through simulation. The ion-migration-induced force was estimated at $\sim 6.16 \times 10^{-10}$ N, whereas the electrostatic force was $\sim 2.5 \times 10^{-15}$ N - five orders of magnitude smaller. Thus, electrostatic contributions can be ignored relative to the force generated by ion migration.

Accordingly, electrostatic force is not included in our current explanation of the hydrogel actuation mechanism. A figure and caption have been added in the revised supplementary Information (Fig. S32 in the revised supplementary) to clarify this point.

Fig. R13. (Fig. S32 in the revised supplementary). Zeta potential of 30 wt% AAc hydrogel in different solutions. In DI water, the negatively charged hydrogel microcilia bent toward the cathode (Fig. 2A in the revised manuscript). Simulations show that the electrostatic force (2.5×10^{-15} N) is several orders of magnitude smaller than the ion-migration-induced force (6.16×10^{-10} N). Together, the experiments and simulations demonstrate that electrostatic force is negligible in hydrogel microcilia actuation.

Specific Comment 2:

2. Most of the results, including those of the flow pattern generation, concern symmetrically rotating microcilia leading to vortical motions. To create net flow, asymmetric motions must be achieved. Can this be done with this technology? It seems relatively straightforward to do by applying different voltages to the electrodes surrounding the cilia, so that a time-dependent rotating electrical field magnitude is achieved causing a tilted conical motion. Can this be demonstrated?

Response:

We would like to clarify a potential misunderstanding regarding the symmetry and reciprocity of our hydrogel microcilia motions. **The rotating motion demonstrated in this work is inherently non-reciprocal, as it breaks time-reversal symmetry (21).** It should also be noted that mouse ventral nodal cilia generate flow through rotational motion, which plays a critical role in embryonic development (Fig. R2 (Fig. S5 in the revised supplementary), Movie S1) (8). To illustrate this clearly, we compared the non-reciprocal rotation of hydrogel microcilia (Fig. R14B) with a typical reciprocal scallop-like motion (Fig. R14A). In the scallop case, the reversed sequence ($0^\circ-30^\circ-0^\circ$) precisely replicates the forward motion, indicating reciprocity. By contrast, reversing the rotation of our microcilia from clockwise ($0^\circ-270^\circ-180^\circ-90^\circ-0^\circ$) to counterclockwise ($0^\circ-90^\circ-180^\circ-270^\circ-0^\circ$) results in distinct and non-overlapping motion sequences. This explicitly

demonstrates the non-reciprocal nature of the hydrogel microcilia rotation, which enables net fluid transport even at low Reynolds numbers.

Experimental evidence is presented in the original Fig. 4, where diverse net flow patterns—such as L-shaped flows (Fig. 4E) and upward and downward directional flows (Fig. 4F)—are observed. Such net fluid transport would not arise if the motions were symmetric or reciprocal at the microscale.

In addition, the reviewer suggested exploring tilted or conical motions. These are indeed feasible within our system and further expand its capabilities. By tuning the duty cycle of the applied square-wave voltage, we successfully realized tilted rotations, as shown in Fig. R15 and Movie S3. This further demonstrates that our hydrogel cilia possess the ability to reconfigure their motion in a manner analogous to biological cilia (Fig. R2 (Fig. S5 in the revised supplementary)) (8)—a feature that has not been reported for existing artificial cilia (please refer to our response to Major Comment 1).

We have added Fig. R15 (Fig. S6 in the revised supplementary) and Movie S3 in the revised manuscript to highlight the motion reconfigurability of our hydrogel microcilia.

Fig. R14. Reciprocal scallop motion and non-reciprocal hydrogel microcilia rotation.

Fig. R15 (Fig. S6 in the revised supplementary). Reconfigurable motion by adjusting the control signal. The dashed lines show the motion trajectories and the right column are the corresponding control signals.

Specific Comment 3:

3. In relation to this, the effect of the electrical field on the motion of the cilia is not quantified, e.g. in Fig. 2 – why not? This seems to be a control parameter that can be used in practice. Throughout the manuscript, only a few field magnitudes are used (but not compared).

Response:

In fact, we have already quantified the effect of varying electric field strengths on cilia motion, as shown in Fig. R16 (also the supplementary Fig. S20 in the original manuscript). These results demonstrate that increasing the electric field strength enhances the bending amplitude and overall performance of the hydrogel microcilia.

This information may not have been sufficiently emphasized or clearly cross-referenced in the main manuscript. To improve clarity and readability, we have revised the manuscript to explicitly reference the corresponding supplementary figures in the discussion of electric field effects (Section Characterization on dynamics).

In addition to diameter, higher electric field strength also accelerates ion transport and thereby enhances the actuator's performance (fig. S17).

Fig. R16 (Supplementary Fig. S20 in the original manuscript). Influence of the electric field intensity on the hydrogel cilia dynamic performance. A smaller electrode distance can enhance the actuation electric field under the same applied electric field. A 1.5 V voltage is applied on electrodes with a distance between 75-230 μm to generate the corresponding electric field intensity. Hydrogel cilia parameters: diameter 10 μm , height 90 μm , AAc 30 wt%. Working environments: DI water.

Specific Comment 4:

4. The speed of the microcilia motion is only analyzed for a step response (Fig. 2); how about oscillations? Up to which frequency can the microcilia still follow the electrical field imposed and by which effect is this limited?

Response:

In addition to the step responses shown in the original Fig. 2A and Fig. 2B, we indeed conducted systematic experiments to characterize the frequency response of the hydrogel microcilia, as reported in the original manuscript under Section “**Characterization on dynamics**” (Fig. R17, originally Fig. 2C). These experiments examined actuators with different hydrogel compositions, assessed durability under dynamic signals, and analyzed the frequency response of actuators with varying diameters and in different solutions.

Based on the classic -3 dB criterion, the effective dynamic response bandwidth is defined as the frequency at which the response amplitude decreases to 70.7% of the low-frequency (5 Hz) response, which corresponds to the maximum frequency at which the microcilia can reliably follow the applied signal. The cutoff frequency data are presented in Fig. R18. A range is reported rather than a single cutoff value because frequency-domain analysis strictly applies to linear time-invariant systems with sinusoidal inputs. In our case, the actuators are driven by square-wave signals, and the dynamic response was tested at several discrete frequencies. As our current control system cannot generate sinusoidal inputs suitable for precise frequency-domain analysis, we provide an approximate cutoff frequency range instead of an exact value.

The dimensionless Sperm number (Sp) is widely used to describe the reduction in bending at higher frequencies in low Reynolds number environments ($Re \approx 0.001$ in our system). As frequency increases, Sp also increases, indicating that viscous effects become more dominant, leading to faster decay of bending along the actuator length and reduced tip displacement. Further details are provided in our response to Reviewer 3's Specific Comment 4.

Fig. R17. Dynamic test of the hydrogel cilia.

Fig. R18. Cutoff frequency data.

Specific Comment 5:

5. In the discussion of the combined effect of actuation frequency and microcilia size (Fig. 2Ciii), the influence of bending stiffness, inertia, and fluid drag is put forward. First, do inertial effects

really play a role at all? A quick calculation of Reynolds number results in values smaller than 1 in all cases, which suggests that fluid flow inertial effects are absent. Second, the difference in bending between the two cilia sizes must be the result of the actuation force (ion motion) scaling differently with size than the counteracting forces (elasticity, viscosity) – it cannot be explained by considering only separate forces. How does that work?

Response:

The reviewer is correct. Our system operates at $Re \approx 0.001 \ll 1$, where inertial effects are negligible. We therefore analyze the dominant actuation force arising from ion migration and the counteracting elastic and viscous forces at different length scales ($2 \mu\text{m}$ vs. $10 \mu\text{m}$). Each cilium can be approximated as a pillar of radius a and length L , with a fixed aspect ratio $AR = L/a = 18$. To simplify the scaling analysis, we introduce an overall size parameter s , representing the characteristic geometric scale of the cilium. Both the radius (a) and length (L) scale linearly with s (i.e., $a \propto s, L \propto s$), while the aspect ratio ($AR = L/a$) remains constant.

Actuation force (ion migration).

A closed-form scaling law is difficult to derive directly due to the complexity of the coupled Poisson–Nernst–Planck equations (see **supplementary “Coupled electro-chemo-mechanical model”** for details). Instead, we use results from our fully coupled simulations to infer scaling. For step response simulation in DI water:

- $d = 2 \mu\text{m}$ ($L = 18 \mu\text{m}$): tip displacement $y \approx 3.7 \mu\text{m}$, actuation force $F_{actuation} \approx 5.65 \times 10^{-12}$ N.
- $d = 10 \mu\text{m}$ ($L = 90 \mu\text{m}$): tip displacement $y \approx 18.3 \mu\text{m}$, actuation force $F_{actuation} \approx 2.1 \times 10^{-10}$ N.

The force ratio $\frac{2.1 \times 10^{-10} \text{ N}}{5.65 \times 10^{-12} \text{ N}} = 37$ is close to s^2 for a five-fold increase in size, suggesting that the actuation force scales as $F_{actuation} \sim s^2$.

Viscous resistance

For small-amplitude bending of a slender filament in Stokes flow (22), the transverse drag per unit length is

$$\zeta_{\perp} \approx \frac{4\pi\mu}{\ln(L/a) + 1/2}$$

which is size-independent here because $L/a=AR$ is fixed. For a small angular velocity Ω about the base, the total transverse viscous force scales as

$$F_{\perp} = \int_0^L \zeta_{\perp} \Omega x dx = \frac{1}{2} \zeta_{\perp} \Omega L^2 \Rightarrow F_{\perp} \sim s^2$$

Elastic restoring force

Approximating the ion-driven loading as uniformly distributed along the cilia, the tip-deflection relation for a cantilever gives

$$F_{\text{elastic}} = \frac{8EI}{L^3} y$$

with $I = \pi a^4/4$ and $L=AR \times a$. Hence

$$F_{\text{elastic}} = \frac{2E\pi a}{AR^3} y \Rightarrow F_{\text{elastic}} \sim sy$$

Scaling of displacement

From force balance $F_{\text{actuation}} = F_{\perp} + F_{\text{elastic}}$, we obtain

$$y = \frac{F_{\text{elastic}} \times AR^3}{2E\pi a} = \frac{(F_{\text{actuation}} - F_{\perp}) \times AR^3}{2E\pi a} \sim \frac{s^2}{s} \sim s$$

Therefore, tip displacement scales linearly with size, while bending angle ($\theta \sim y/L$) remains nearly scale-independent. This is confirmed by simulations, where the 2 μm tip displacement ($y \approx 3.7 \mu\text{m}$) is $\sim 1/5$ that of the 10 μm cilium ($y \approx 18.3 \mu\text{m}$), and by experiments at 5 Hz, where both show similar bending angles (Fig. R19).

Frequency dependence

At higher frequencies, however, the 10 μm cilium exhibits reduced bending compared to the 2 μm cilium. This can be understood as follows: at low frequency, both cilia reach a “saturated” bending state since ions have sufficient time to redistribute. At higher frequencies, the shorter ion transport path in the 2 μm cilium allows faster response, leading to greater dynamic bending. Step-response simulation sequences (Fig. R20) confirm this behavior: within the same time frame, the 2 μm cilium reaches its maximum bending angle earlier, indicating a faster electromechanical response compared to the 10 μm cilium.

The above analysis was added in the revised supplementary ‘Analysis between the 2 μm - and 10 μm -diameter hydrogel cilia’

Fig. R19. Varying diameter test.

Fig. R20 (Fig. S31 in the revised supplementary). Simulation comparison between the 2 μm and 10 μm hydrogel cilia.

Specific Comment 6:

6. Fig. 4 shows flow patterns generated by actuated arrays of microcilia, but the flow is not quantified. Since PIV measurements were carried out, the magnitude of the fluid flow velocities should be known. Why are these not reported? These are interesting for practical applications.

Response:

We have revised Fig. 4 (Fig. R21) to include normalized velocity in the simulation results. Since the simulations are two-dimensional, absolute velocity differs from three-dimensional experiments, but the relative magnitudes and directions reliably capture flow structures and velocity distributions.

We also added the magnitude of the fluid flow velocities to the PIV results. It should be noted that the patterned electrodes on the substrate influence the accuracy of PIV analysis. For cases A–C, where the electrode density is relatively low, the PIV results are reliable. For cases D–G, the electrode density is higher, and the trajectories of the PIV tracer particles overlap substantially with the underlying electrodes, leading to larger errors in velocity quantification. From the particle imaging velocimetry (PIV) results, we estimate that the maximum flow velocities generated by cilia in cases A–D range between 50 and 250 μm/s. In cases F–G, particle tracking measurements indicate velocities of approximately 55 μm/s, 18 μm/s, and 22 μm/s, respectively.

The above highlighted contents were added in the supplementary ‘Particle Image Velocimetry (PIV) analysis’.

Fig. R21 (Fig. 4 in the revised manuscript). Dynamic fluid manipulation by gel microcilia arrays.

Specific Comment 7:

7. The abstract mentions that the presented microcilia enable “particle transportation”. Is this really shown in the manuscript? Fig. 4, last column, shows particle tracking images, but these are particles that follow the fluid flow, aren't they? If so, the manuscript does not show direct particle transportation, but particles taken by fluid flow; any system that can generate fluid flow can cause such particle transportation and mentioning “particle transportation” in the abstract seems superfluous and sets expectations that are not met.

Response

We would like to emphasize that in both biological and microfluidic fields, the use of flow to transport cells or particles is a well-established and widely accepted description. For example, in the biological literature it has been noted that “the leftward flow will efficiently transport small membrane carriers to the left like the plastic beads” (8). In microfluidics, the term “particle transportation” is similarly used to denote the movement of particles driven by controlled fluid flows, and it is commonly applied in studies on particle or cell manipulation, separation, trapping, and directed transport (23-28). Therefore, our use of the term is chosen to be consistent with established conventions in both fields.

Specific Comment 8:

8. Fig. 4 shows interesting fluid patterns. What would be applications of such flows?

Response

In nature, biological cilia perform diverse functions by manipulating surrounding fluids, including airway clearance (29), gamete transport (30), and regulation of embryonic development (8). As shown in Fig. R21 (Fig. 4 in the revised manuscript), our hydrogel cilia similarly enable controllable fluid transport and directional particle motion, suggesting potential for future application-oriented studies. We also conducted new experiments actuating the cilia in human saliva, human serum, and mouse plasma. The results demonstrate their potential for mixing physiological fluids. This discussion has been added to the “Characterization on dynamics” section of the revised manuscript.

To further illustrate this, we developed a biomimetic system inspired by star larvae, which generate complex vortex arrays through their cilia to support feeding and other essential processes (17). The active boundary conditions arising from cilia–environment interactions may have potential effects in both biological and synthetic systems (17). Existing studies of such phenomena have been largely qualitative, relying on biological observation and lacking systematic

quantification (17, 31). To address this gap, we created an artificial star larvae platform by combining microscale 3D printing of the larval body, microelectrode patterning on its curved surface, and integration of hydrogel cilia. As shown in Fig. R22 (Movie S21), the artificial star larvae reproduce biologically analogous vortex arrays under electrical control, providing a reliable robotic platform for quantitative investigation of biomimetic processes and active boundary conditions.

Beyond mimicking biological cilia, we also highlight the versatility of the hydrogel actuators. By tuning processing parameters, we patterned hydrogel actuators and microelectrodes onto complex 3D curved surfaces, achieving motion and flow manipulation directly on such geometries (Fig. R4A, Movies S17-18). The actuators can further be integrated into 3D lattice structures (e.g., pyramid frameworks) or directly fabricated as intrinsically 3D helical structures (Fig. R4Ci-ii, Movies S19-20). Finally, we demonstrate the integration of hydrogel actuators with micromechanical structures (Fig. R4Ciii-iv, Movies S22-23). Under electrical stimulation, hydrogel bending was transduced into rotary and flapping motions via mechanical linkages, thereby broadening the application space of these actuators. This capability underscores the potential of hydrogel actuators to interface with complex micromachine designs, enriching actuation strategies and advancing the field of microscale robotics.

Fig. R22. Biomimetic artificial star larvae. (A) Photo of artificial star larvae with patterned electrodes prior to cilia integration. (B) SEM image of larvae with distributed cilia; arrows indicate rotation directions. (C) PIV result of the flow field generated by cilia motion. (D) Schematic showing the flow generated by cilia motion in biological star larvae (17).

Reviewer #2

Overall Comment:

I co-reviewed this manuscript with one of the reviewers who provided the listed reports.

Response:

We thank the reviewer for taking time to evaluate our manuscript. Please check our point-by-point responses above and below.

Reviewer #3

Overall Comment:

The manuscript discusses a ciliary platform using ion mitigation inside a hydrogel to instigate bending motions. Thorough insights are given on the mechanisms that drives this bending deformation, detailing the performance in DI and saline water. Carpets of cilia of different size and amount are fabricated that are either individually controlled or sectionally controlled.

The capabilities of such cilia arrays have somewhat being demonstrated by means of imposing flow patterns.

Response:

We thank the reviewer for spending time reviewing our manuscript. Please check our detailed response below.

Major Comment 1

1. To my feeling, the innovation of this manuscript lies in the fabrication process, creating cilia of micrometer dimensions, that span centimeter scaled surfaces. However the details of this process are being given in SI.

In contrast, the manuscript details on (i) differential swelling hydrogels, where basic mechanisms have already been detailed in literature (44-48); (ii) patches of cilia that show analogous modalities as displayed in (37). And although the results are well described, it is not where the real innovation lies.

Response:

Thank you for this insightful suggestion. In our revision, we considered this comment very carefully and **fully agree that the main contribution of this work lies in the actuation mechanism and its realization – direct miniature hydrogel actuator printing technique**. The three points highlighted by the reviewer, together with the newly added demonstration (Fig. R4), exemplify this actuation mechanism and the associated fabrication approach.

Details can be found in our **Response to the Editor** at the beginning, as well as in our replies to **Reviewer 1's Major Comments 1 and 2**, and **Specific Comment 8**.

Major Comment 2

2. Further, the manuscript lacks a clear demonstrator for such cilia.

Response:

After clarifying our contribution, we believe the cilia and the newly added demonstrations are demonstrator of our proposed low-voltage driven hydrogel actuation mechanism and its associated fabrication approach.

Please refer to our Response to the Editor at the beginning and our response to Reviewer 1's Specific Comment 8 for details on the application scenarios of the hydrogel cilia and hydrogel actuator.

Specific Comment 1

Other critical questions:

3. In the introduction, it is stated that current cilia are incompatible with ionic solutions (line 73). However the presented solution is also highly dependent on the surrounding fluid. In what sense is this solution capable of driving a wide variety of fluids (e.g. blood, saliva, urine, etc...). As such, the conclusion on line 240-241 seems to not well funded.

Response:

The artificial cilia referred to in the original manuscript line 73 are DEA cilia, which operate at relatively high voltages, typically ranging from several tens to over one hundred volts (7). Such high voltages restrict their operation to insulating media like silicone oil, as water or other aqueous ionic solutions would undergo electrolysis under these conditions.

By contrast, the hydrogel cilia developed in this work are driven at a low voltage of 1.5 V and have demonstrated stable performance in DI water, physiological saline, and DPBS solutions. To further support the conclusion presented in the original manuscript lines 240–241 and following the reviewer's suggestion, we additionally tested our hydrogel cilia in several biofluids, including human saliva, human serum, and mouse plasma (Fig. R23). In all cases, the hydrogel cilia exhibited stable actuation performance. Although performance was moderately reduced in more complex ionic environments, coordinated array motions of small bending actuators were still able to generate controllable flows (1).

Fig. R23. Hydrogel cilia working in different solutions

Specific Comment 2

4. In line 77 it is stated that soft substrates are beneficial, however throughout the manuscript, the softness of the substrate is not harnessed in applications.

Response:

We thank the reviewer for raising this point. To demonstrate the flexibility of our system, we integrated the polyimide substrate, microelectrodes, and hydrogel actuators onto a three-dimensional curved surface with a height variation of 100 μm (Fig. R24, Movies S17-18). The hydrogel actuators exhibited stable motion and effective fluid manipulation on this curved geometry. As shown in Fig. R24, coordinated hydrogel motion enabled controllable flows along both right-to-left (Fig. R24A) and top-to-bottom (Fig. R24B) directions on the 3D surface.

Therefore, the softness of the substrate allows cilia to be fabricated on three-dimensional surfaces. The hill-shaped surface more closely resembles the non-flat morphology of physiological substrates and also provides engineering opportunities beyond those offered by previously reported approaches (Fig. R1).

Fig. R24 (Fig. S25 in the revised supplementary). Cilia patterned on a 3D curved surface. (A) Right-to-left flow on 3D surface. Left: SEM image with arrows indicating cilia rotation; Right: PIV result with the flow region highlighted by a red dashed box. (B) Top-to-bottom flow on 3D surface. Left: SEM image with arrows indicating rotation directions; Right: PIV result with the flow region marked by a red dashed box.

Specific Comment 3

5. Is there an intuitive explanation for the fact that higher AAC concentrations produce smaller H⁺ ion concentration growth rate, besides simulation (line 140)?

Response:

Beyond the ion concentration simulation trends, the reduced H⁺ growth rate at higher AAC concentrations can be intuitively explained by fixed-charge (Donnan) electrostatics and acid–base buffering within the gel. An increase in AAC concentration raises the density of negatively charged –COO[–] groups, which (i) renders the gel interior more negatively charged, pulling H⁺ ions closer and reducing their effective transport rate, and (ii) the additional –COO[–] groups temporarily bind incoming H⁺ ions ($-\text{COO}^- + \text{H}^+ \rightleftharpoons -\text{COOH}$), thereby slowing the net increase of free H⁺ ions inside the gel. Within the Nernst–Planck framework, this leads to a reduced net H⁺ flux and consequently a slower accumulation rate of free H⁺ inside the gel.

To test this mechanism, we performed simulations to evaluate the internal potential generated by fixed -COO^- groups, which exerts a strong electrostatic attraction on H^+ ions and slows their migration. As shown in Fig. R25, the internal potential generated by fixed -COO^- groups increases monotonically with AAc concentration, consistent with higher fixed-charge density and stronger H^+ attraction/binding. In addition to this explanation, we also conducted fully coupled electro-chemo-mechanical simulations. As shown in Fig. R26, the simulation results agree with the experimental trends, confirming that the bending amplitude of the hydrogel actuator decreases with increasing AAc concentration. Full details of the modeling are provided in the **supplementary “Coupled electro-chemo-mechanical model”** section.

Figure R25. Mean internal potential of hydrogels as a function of AAc concentration.

Fig. R26. Experimental and fully-coupled simulations of hydrogel bending in DI water.

6. Regarding the reduced stroke at higher frequency, the dimensionless sperm number is typically used to describe this phenomena.

Response:

Thanks for the suggestion. The Sperm number (Sp) characterizes the balance between viscous drag and bending elasticity in low Reynolds number environments ($Re \approx 0.001$ in our system). We calculated Sp for the 30 wt% AAc hydrogel cilia used in this study (diameter $10 \mu\text{m}$, height $90 \mu\text{m}$, Young's modulus 1314 Pa) under different actuation frequencies. As shown in Fig. R27, Sp is ~ 1.8 at 5 Hz and increases with frequency, reaching ~ 3.3 at 50 Hz . This indicates that viscous effects become more pronounced at higher frequencies, causing faster decay of bending along the actuator length and reduced tip displacement. These results are consistent with our experimental observations (Fig. 2C in the revised manuscript) that bending amplitude decreases with increasing frequency.

The above discussion about sperm number was added in the revised manuscript 'Characterization on dynamics'

Fig. R27 (Fig. S16 in the revised manuscript). Sperm number of 30 wt% AAc hydrogel cilia at different actuation frequencies. The hydrogel has an aspect ratio of 9.

7. In SI movie 2, it seems that the movie in DI water is cut short, as it seems to bend backwards towards the anode. Can you provide a longer movie, and explain if this is happening?

Response:

We have provided a complete video of the step response in DI water (Movie S7), including the initial state, application of the voltage signal, maintenance of the voltage signal, and removal of the voltage signal. Each stage is clearly annotated. During the period when the voltage signal is maintained, the hydrogel actuator continues to bend toward the cathode, with only a very slight backward motion. This effect is likely due to minor signal instabilities in the control board during

step response generation, where small voltage fluctuations can influence hydrogel motion in DI water because of the rapid mobility of H^+ ions. Our control system is optimized for dynamic square-wave signals, and in step mode—where a constant output must be sustained—some instability in signal delivery can occur.

Specific Comment 6

8. The manuscript has grammatical mistakes (e.g. line 113 misses a verb), and should be revised in this sense.

Response:

We thank the reviewer for pointing out the grammatical mistakes. These issues have been corrected, and we have carefully examined the entire manuscript to further minimize grammatical issues.

Specific Comment 7

9. A formal conclusion is lacking.

Response

We have added the following conclusion in the revised manuscript 'Discussion':

By leveraging two-photon polymerization to engineer nanometer-scale porosity, we enhance ion transport and electroosmotic flow, enabling fast, large-amplitude, and independently programmable motions. Systematic studies reveal that this low-voltage actuation mechanism is unique to the microscale, where ionic hydrogels operate through principles fundamentally different from their millimeter-scale counterparts. The resulting hydrogel microcilia faithfully reproduce key features of biological ciliary motion while supporting versatile flow control, biomimetic model systems, and integration with micromachines. Overall, our approach enables direct printing of miniature hydrogel actuators that deliver superior speed, strength, and functional versatility compared to existing systems.

Reviewer #4

Overall Comment

The authors present a scalable, high-precision, and electrically controllable soft-body ciliary array system for microscale fluid manipulation. The work is particularly impressive in demonstrating programmable 3D motion with fast response times and independent actuation of individual cilia. The concept is well-articulated, and the manuscript is written clearly with a broad audience in mind. I find the study to be both innovative and significant. I recommend major revision to strengthen the manuscript before publication. Please see my comments below:

Response:

We thank the reviewer for the recognition and support of our work. We have addressed the specific comment as follows.

Specific Comment 1

1. To fully demonstrate the programmability and robustness of the ciliary system, additional quantitative characterization is needed. While the effects of NaCl concentration, cilia diameter, and solution properties are examined, the influence of electric field intensity and cilia height on actuation behavior should be studied.

Response:

We have conducted a new experiment to study the effect of electric field strength on cilia actuation. As shown in Fig. R28, hydrogel cilia performance improved with increasing field strength. Microelectrodes with different spacing were used to generate varying electric fields, while the applied voltage was maintained at 1.5 V.

We have conducted a new experiment to examine the influence of cilia height. The tested cilia had a diameter of 10 μm and heights of 90 μm , 70 μm , and 50 μm . As shown in Fig. R29, taller cilia displayed greater bending angles. For example, under the same electric signal, a 90 μm -high cilium bent more than a 70 μm -high cilium because the additional 20 μm length was also actuated by the field, contributing to the overall deformation.

Fig. R28 (Fig. S17 in the revised supplementary). Influence of the electric field intensity on the hydrogel cilia dynamic performance. A smaller electrode distance can enhance the actuation electric field under the same applied electric field. A 1.5 V voltage is applied on electrodes with a distance between 75-230 μm to generate the corresponding electric field intensity. Hydrogel cilia parameters: diameter 10 μm , height 90 μm , AAc 30 wt%. Working environments: DI water.

Fig. R29 (Fig. S18 in the revised supplementary). Influence of the hydrogel cilia height on their dynamic performance. The tested cilia had a diameter of 10 μm and heights of 90 μm , 70 μm , and 50 μm .

Specific Comment 2

2. The authors should elaborate on potential applications of the system. Providing at least one concrete example (e.g., microfluidic mixing, targeted cargo transport, or biomedical sensing) would highlight the broader impact of the technology.

Response:

Due to the innovation on using TPP to create nanometer-scale pores in the hydrogel network, the actuation performance achieved in this work is substantially enhanced compared with previous studies (Table R1). Our approach functions across a wide range of fluids, from DI water to various physiological fluids (Fig. 2Civ in the revised manuscript, Movie S6), and can therefore be applied to mixing these fluid in addition to the fluid manipulation demonstrated in the manuscript (Fig. 4 in the revised manuscript, Fig. R21).

Since the actuation mechanism and the associated printing technique constitute the main contribution of this work (**Response to Editor**), we further emphasize this point by adding new demonstrations in the revision. Specifically, we performed additional experiments showing their operation on complex three-dimensional curved surfaces, their potential application in constructing artificial microorganisms for studying biological processes and propelling microrobots, and their integration with other three-dimensional structures, including micromachines. These demonstrations highlight the multifunctionality of our hydrogel actuators in microscale actuation (Fig. R4).

For details, please refer to the summary at the beginning, our response to **Reviewer 1's Specific Comment 8**, and the **Response to the Editor** at the beginning regarding application scenarios of the hydrogel cilia and hydrogel microactuators.

Specific Comment 3

3. Can the authors provide data (experimental or simulated) on the mechanical strength of the hydrogel cilia and estimate the magnitude of the electric field-induced forces? This would clarify the relationship between applied field and resulting displacement.

Response:

We measured the Young's modulus of AAc hydrogels at 15, 30, 45, and 60 wt% by AFM. As shown in Fig. R30, the modulus increases from ~1.2 kPa to ~1.5 kPa with increasing AAc content.

To estimate the electric-field-induced force, we consider the 30 wt% AAc hydrogel cilia (diameter 10 μm , length 90 μm). In DI water, the fully coupled simulation predicts bending toward the cathode with a tip lateral displacement of ~18.3 μm and a net driving force of $\sim 2.1 \times 10^{-10}$ N (Fig. R31). To cross-check this value, we approximated the cilium as a cantilever beam under a uniform distributed load, since ion migration generates force along the entire cilia height. The bending stiffness is $B = EI \approx 6.46 \times 10^{-19}$ N·m² (with $E \approx 1.3$ kPa, and $I = \pi r^4/4$, $r = 5$ μm). For a maximum tip deflection y , the load is $F = 8By/L^3$. Using $y = 18.3$ μm and $L = 90$ μm gives $F \approx 1.3 \times 10^{-10}$ N in good agreement with the simulation. Together, these analyses indicate that the electric-field-induced force on the 10 μm hydrogel cilia is on the order of 10^{-10} N.

In fact, this shows another similarities of our hydrogel cilia with biological cilia as the force provided by the biological cilia are from tens of pN in human motile cilia (32) to hundreds of pN in sperm tail (33). And this exactly matches what our cilia can provide.

Fig. R30 (Fig. S7 in the revised manuscript). Young's modulus of hydrogels with different AAC ratios.

Fig. R31. Hydrogel cilia's simulation in DI water.

Specific Comment 4

4. Given the high electric field used, please address whether water ionization or electrolysis occurs under experimental conditions. This could have implications for long-term stability or application in biological systems.

Response:

Thanks for raising this up. In fact, this is one critical advantage of proposed actuator in compared with previous work (7) using high voltages typically ranging from several tens to over one hundred volts. Such high voltages restrict their operation to insulating media like silicone oil, as water or other aqueous ionic solutions would undergo electrolysis under these conditions. Moreover, previously reported centimeter- to millimeter-scale hydrogel actuators generally required tens of volts for operation in aqueous solutions, and their results showed pronounced electrolysis (12, 34), severely limiting their long-term practical applicability.

In the current case, although the local electric field across the microelectrodes is high ($\sim 15,000$ V/m for a $100\ \mu\text{m}$ gap), **the applied potential difference is only 1.5 V. This is below the threshold for sustained water electrolysis in aqueous media ($>1.5\text{--}2$ V when overpotentials are considered) (35), and no gas evolution was observed experimentally.** Therefore, electrolysis does not occur under our operating conditions, ensuring stability and biocompatibility.

The actuation mechanism of the hydrogel cilia is governed solely by ion migration under the applied field, without electrochemical reactions. To evaluate durability, we tested the hydrogel cilia for 330,000 cycles (~ 5 hours at 20 Hz). As shown in Fig. R32, stable actuation was maintained throughout. The bending amplitude decreased to $\sim 70\%$ of the initial value after $\sim 80,000$ cycles but then stabilized without further degradation, demonstrating the potential of the hydrogel cilia for long-term operation.

Fig. R32. Hydrogel cilia durability test.

Specific Comment 5

5. The definition of bending angle should be clarified. Please specify how bending is quantified in this work.

Response:

As shown in Fig. R33, we measured the lateral displacement as the horizontal distance between the left and right amplitude positions of the hydrogel tip during bending. The bending angle was then calculated under the assumption that (i) the hydrogel bends uniformly and (ii) the neutral axis of the hydrogel body (red dashed line) forms a circular arc after deformation. This assumption, combined with the geometric relationship illustrated in Fig. R33, provides the basis for the angle calculation. Furthermore, our fully coupled simulations validated these assumptions, showing that the hydrogel exhibits uniform bending under the applied electric field (Fig. R12).

Fig. R33. Schematic to show the definition of bending angle.

Specific Comment 6

6. Reduced bending amplitudes may also be influenced by the Young's modulus of the hydrogel. Please discuss whether variations in polymer crosslinking or photo-initiator concentration affect mechanical stiffness, and how that might influence actuation amplitude.

Response:

The reviewer is correct. For UV-cured hydrogels, both exposure intensity and photo-initiator concentration affect material stiffness: higher values of either parameter result in stiffer hydrogels (36, 37). To verify this, we fabricated hydrogel cilia (10 μm diameter, 90 μm height) under different printing powers (17.5, 22.5, and 27.5 mW) and with varying photo-initiator concentrations (7.5, 15, and 22.5 wt%).

As shown in Fig. R34A and Fig. R34B, cilia produced at lower printing power or with lower photo-initiator content exhibited larger bending amplitudes, confirming reduced stiffness and improved actuation performance. Within a practical range, lowering these parameters can enhance performance, but excessively low values may cause incomplete polymerization and hinder fabrication of high-aspect-ratio structures. Excessive power or initiator concentration increases network density and further closes the pores, which can also impair actuation performance.

Fig. R34 (Fig. S19 in the revised supplementary). Hydrogel cilia dynamic test with different printing power and photo initiator concentration.

Specific Comment 7

7. The mechanism by which the hydrogel responds to the electric field requires more explanation. A deeper discussion—supported by references and your own characterizations—would benefit readers unfamiliar with electroactive gels.

Response:

Thanks for the reviewer’s suggestions. We have conducted systematic control experiments and developed a fully coupled electro-chemo-mechanical model. For details, please refer to our response to Reviewer 1’s Major Comment 2.

In summary, millimeter-scale hydrogels bend through bath-induced pH or interfacial osmotic gradients, where electrolysis or ion partitioning across the gel–solution interface dominates. In contrast, micrometer-scale hydrogels rely on internal ion migration and nanometer-scale channels, producing opposite bending directions and responses that are over two orders of magnitude faster.

In this model, the influence of densely distributed $-\text{COO}^-$ groups on ion migration within the hydrogel is also included, capturing the influence of the nanometer-scale hydrogel pores (TEM results in Fig. R3Bii).

The simulation explains our experiment results well. Comprehensive descriptions of the mechanism, modeling, and simulations are also provided in the **supplementary “Coupled electro-chemo-mechanical model”** section.

Specific Comment 8

8. A mechanical model or analysis of the deformation behavior would significantly enhance the study. Even a simplified analysis could provide valuable insights into the bending mechanism and design parameters.

Response:

A fully coupled electro-chemo-mechanical model and corresponding simulations were developed in this revision. Please refer to our response to **Reviewer 1's Major Comment 2**, with full details of the mechanism, modeling, and simulations provided in the **supplementary "Coupled electro-chemo-mechanical model"** section.

Specific Comment 9

9. Please clarify the polarity and direction of the applied field in Fig. 1C, and revise the region labeling (Region 1 vs. Region 2) to avoid confusion. A clearer schematic, as seen in Supplementary Fig. S13, would be helpful here.

Response:

As shown in Fig. R35 (Fig. 1 in the revised manuscript), we have added the electrode polarity in panel C and relocated the region labeling to the bottom of the schematic to improve clarity.

Fig. R35 (Fig. 1 in the revised manuscript). Overall concept.

Specific Comment 10

10. If AAc content plays a key role, why do the bending angles in Fig. 2C show minimal variation across groups at higher frequencies (e.g., 20–50 Hz)? Statistical analysis is needed to confirm trends. Additionally, can the authors provide Young’s modulus measurements for cilia of different diameters?

Response:

We used the Sperm number (Sp) to explain the reduced bending amplitude of hydrogel cilia at higher frequencies (please refer to our response to **Reviewer 3’s Specific Comment 6**). For cilia with different AAc concentrations, Sp increases with frequency, leading to decreased bending angles across all groups and smaller absolute differences at higher frequencies, reducing the variation.

To further verify that hydrogel cilia with different AAc concentrations exhibit statistically significant differences and consistent trends, we performed Analysis of Variance (ANOVA) and Jonckheere’s trend test. As shown in Table R2, for all tested frequencies, the F-statistic values were much greater than 1 and all p-values were well below 0.05, indicating significant differences in bending among groups. Jonckheere’s trend test (Table R3) further showed negative standardized test statistics with p-values well below 0.05 at all frequencies, confirming a significant monotonic decrease in bending amplitude with increasing AAc concentration.

Direct AFM test on individual 2–10 μm cilia is not feasible in our setup because reliable nano-mechanical probing requires a round tip ($\sim 5 \mu\text{m}$ diameter in our system) and a contact area larger than the probe to avoid edge effects and allow multi-site sampling. Instead, we prepared $100 \mu\text{m} \times 100 \mu\text{m} \times 100 \mu\text{m}$ cubic samples using the same 30 wt% AAc solution and the same two-photon printing parameters, and measured a Young’s modulus of $\sim 1314 \text{ Pa}$ (Fig. R36). Given the identical chemistry and processing, we infer that the 2 μm - and 10 μm -diameter cilia fabricated under the same conditions possess essentially the same modulus.

Analysis of variance (ANOVA)

Frequency	F-statistic	p-value
5 Hz	291.8686878	1.13×10^{-6}
10 Hz	103.3282349	2.41×10^{-12}
20 Hz	28.39576218	2.08×10^{-7}
30 Hz	32.26316387	7.37×10^{-8}
40 Hz	19.86127098	3.27×10^{-6}
50 Hz	81.10559624	2.28×10^{-11}

Table R2. ANOVA analysis of the bending data.

Jonckheere's trend test

Frequency	Standardized test statistic	p-value
5 Hz	-4.937707199	3.95×10^{-7}
10 Hz	-3.806149299	7.06×10^{-5}
20 Hz	-4.114755999	1.94×10^{-5}
30 Hz	-4.320493799	7.78×10^{-6}
40 Hz	-4.474797149	3.82×10^{-6}
50 Hz	-4.989141649	3.03×10^{-7}

Table R3. Jonckheere's trend test of the bending data.

Fig. R36 (Fig. S7 in the revised supplementary). Young's modulus of hydrogels with different AAC ratios.

Specific Comment 11

12. A schematic of the electrode configuration used in Fig. 3 would help clarify the observed 180° phase shift and its relation to cilia motion.

Response:

Thanks for the suggestion. As shown in Fig. R37, we have added the electrode polarity to improve clarity.

Fig. R37 (Fig. 3 in the revised manuscript). Dynamic motions of electrically actuated gel microcilia arrays.

Specific Comment 12

13. Figures 2biii–vi show that tip displacement appears insensitive to NaCl concentration, while bending angle is affected. Meanwhile, Fig. 2Civ shows bending angle varies with frequency. Could the authors clarify the relationship between tip displacement and bending angle? Are these quantities independently controlled or correlated?

Response:

Please refer to our response to **Specific Comment 5** regarding the relationship between tip displacement and bending angle. These quantities are independently controlled.

The bending angle data are provided in Fig. 2Civ of the original manuscript, while the tip displacement data are shown in Figs. 2biii–vi (in the original manuscript). These two sets of data were obtained under different experimental conditions and are therefore independent.

For the bending angle measurements (Fig. 2Civ of the original manuscript), the experiments were conducted under a dynamic square-wave signal. Under such conditions, ions undergo repeated acceleration, steady transport, and deceleration.

For the tip displacement measurements (Figs. 2biii–vi of the original manuscript), the experiments were performed using a step input. In this case, ions move unidirectionally, which facilitates clearer identification of the contributions from different ionic species. For example, in a 0.00769 mol/L NaCl solution, the observed transition behavior illustrates the sequential influence of H⁺ and Na⁺ ions on hydrogel bending.

Taken together, these results demonstrate that the two experiments were carried out under entirely different conditions, and their outcomes are independent. The experiments in Figs. 2biii–vi (in the original manuscript) were specifically designed to reveal the bending mechanism of the hydrogel cilia, in which H⁺ ions contract the hydrogel network while Na⁺ ions induce swelling, with their effects competing. In this revision, to more directly illustrate the actuation mechanism, we replaced the data from Figs. 2Biii–x of the original manuscript with results from fully coupled simulations (Figs. R38C–D), which, together with the experimental data in Figs. R38A–B, validate the proposed mechanism.

Fig. R38 (Fig.2A-D in the revised manuscript). Step response of hydrogel microcilia in NaCl solutions of different concentrations. (A) In 0.00769 mol/L NaCl, the gel microcilium first bent toward the cathode and then gradually reversed to bend toward the anode. (B) In 0.1538 mol/L NaCl, the gel microcilium bent only toward the anode. (C–D) Simulated bending in 0.00769 mol/L and 0.1538 mol/L NaCl solutions. Consistent with the experiments, the simulation in 0.00769 mol/L NaCl shows initial bending toward the cathode followed by reversal toward the anode, while in 0.1538 mol/L NaCl the bending occurs only toward the anode. Scale bars: (A) 100 μm , (B) 100 μm .

Specific Comment 13

14. In Figure 4, please include a scale bar to indicate the magnitude of the velocity field.

Response:

Thanks. We have added such information. Please refer to our response to Reviewer 1's Specific Comment 6.

Specific Comment 14

15. The table is incomplete; please include the corresponding fluid flow velocity values.

Response:

Thanks for the reviewer’s suggestion. We have added the fluid flow velocity in the “Fluid manipulation capability” column of Table R4. This table has also been added as Table 1.

Performance index Reported studies	Single cilium size	Maximum cilia numbers shown in the work	Repro-grammable coordinated motions	Motion frequency	Fluid manipulation capability	Durability test	Substrate
Pneumatic cilia (26)	cm	6	2D repro-grammable motions	0.25 Hz	Yes (0.5-1 mm/s)	N.A.	Rigid
Magnetic cilia (27-31)	mm to μm	$\sim 10^2$	2D and 3D motions	~ 100 Hz	Yes (0.8 mm/s)	N.A.	Could be built on a stretch-able substrate
Ultrasound-activated cilia (32, 33)	μm	10	No	N.A.	Yes (up to 10 mm/s)	N.A.	Rigid
pH-sensitive cilia (34)	μm	$\sim 10^2$	No	N.A.	N.A.	N.A.	Rigid
Light-responsive cilia (35, 36)	μm	$\sim 10^2$	3D repro-grammable motions	~ 0.1 Hz	N.A.	Tested 100 cycles in (35)	Rigid
Electrostatically actuated cilia (37)	μm	$\sim 10^2$	2D repro-grammable motions	200 Hz	Only in low-dielectric mediums (0.33-2 mm/s)	Tested 600,000 cycles	Rigid
Electrochemically actuated cilia (38, 39)	mm (38), and μm (39)	$\sim 10^2$	2D repro-grammable motions	~ 10 Hz (38), and ~ 100 Hz (39)	Yes (3 mm/s (38), and maximum speed of 60 $\mu\text{m/s}$ (39))	Tested several thousand cycles in the works	Stretch-able substrate (38), rigid substrate (39)
This work	μm	$\sim 10^6$	3D repro-grammable motions	40 Hz	Yes (250 $\mu\text{m/s}$ maximum speed)	Tested 330,000 cycles	Flexible poly-imide substrate

Table R4 (Table 1 in the revised manuscript). Comparison of the proposed gel microcilia with the previously reported works in the literature.

Specific Comment 15

16. Since the article focuses on ciliary actuation, the authors may consider recent developments in ultrasound-driven artificial cilia (<https://doi.org/10.1073/pnas.2418938122>). A brief comparison could help contextualize the present work in relation to other actuation modalities.

Response:

We have added this recently published work as our reference in the introduction part. **This work has also been added to Fig. R1 for a comparison.** Please refer to our Response to the Editor at the beginning.

References

1. W. Wang, Q. Liu, I. Tanasijevic, M. F. Reynolds, A. J. Cortese, M. Z. Miskin, M. C. Cao, D. A. Muller, A. C. Molnar, E. Lauga, P. L. McEuen, I. Cohen, Cilia metasurfaces for electronically programmable microfluidic manipulation. *Nature* **605**, 681-686 (2022).
2. T. Ul Islam, Y. Wang, I. Aggarwal, Z. Cui, H. Eslami Amirabadi, H. Garg, R. Kooi, B. B. Venkataramanachar, T. Wang, S. Zhang, P. R. Onck, J. M. J. den Toonder, Microscopic artificial cilia - a review. *Lab on a Chip* **22**, 1650-1679 (2022).
3. E. Milana, R. Zhang, M. R. Vetrano, S. Peerlinck, M. De Volder, P. R. Onck, D. Reynaerts, B. Gorissen, Metachronal patterns in artificial cilia for low Reynolds number fluid propulsion. *Science Advances* **6**, eabd2508 (2020).
4. P. Amado, C. Dillinger, C. Bahou, A. Hashemi Gheinani, D. Obrist, F. Burkhard, D. Ahmed, F. Clavica, Ultrasound-activated cilia for biofilm control in indwelling medical devices. *Proceedings of the National Academy of Sciences* **122**, e2418938122 (2025).
5. H. Gu, Q. Boehler, H. Cui, E. Secchi, G. Savorana, C. De Marco, S. Gervasoni, Q. Peyron, T. Y. Huang, S. Pane, A. M. Hirt, D. Ahmed, B. J. Nelson, Magnetic cilia carpets with programmable metachronal waves. *Nature Communications* **11**, 2637 (2020).
6. S. Li, M. M. Lerch, J. T. Waters, B. Deng, R. S. Martens, Y. Yao, D. Y. Kim, K. Bertoldi, A. Grinthal, A. C. Balazs, J. Aizenberg, Self-regulated non-reciprocal motions in single-material microstructures. *Nature* **605**, 76-83 (2022).
7. J. Toonder, F. Bos, D. Broer, L. Filippini, M. Gillies, J. de Goede, T. Mol, M. Reijme, W. Talen, H. Wilderbeek, V. Khatavkar, P. Anderson, Artificial cilia for active micro-fluidic mixing. *Lab on a Chip* **8**, 533-541 (2008).
8. Y. Okada, S. Takeda, Y. Tanaka, J. I. Belmonte, N. Hirokawa, Mechanism of nodal flow: a conserved symmetry breaking event in left-right axis determination. *Cell* **121**, 633-644 (2005).
9. T. Shiga, "Deformation and Viscoelastic Behavior of Polymer Gels in Electric Fields" in *Neutron Spin Echo Spectroscopy Viscoelasticity Rheology*. (1997), chap. 2, pp. 131-163.
10. T. Shiga, T. Kurauchi, Deformation of polyelectrolyte gels under the influence of electric field. *Journal of Applied Polymer Science* **39**, 2305-2320 (2003).
11. M. Doi, M. Matsumoto, Y. Hirose, Deformation of ionic polymer gels by electric fields. *Macromolecules* **25**, 5504-5511 (2002).
12. J. Zhang, J. Liao, Z. Liu, R. Zhang, M. Sitti, Liquid Metal Microdroplet-Initiated Ultra-Fast Polymerization of a Stimuli-Responsive Hydrogel Composite. *Advanced Functional Materials* **34**, 2308238 (2023).
13. S. J. Bryant, J. L. Cuy, K. D. Hauch, B. D. Ratner, Photo-patterning of porous hydrogels for tissue engineering. *Biomaterials* **28**, 2978-2986 (2007).
14. S. Yao, J. G. Santiago, Porous glass electroosmotic pumps: theory. *Journal of Colloid and Interface Science* **268**, 133-142 (2003).
15. S. Yao, D. E. Hertzog, S. Zeng, J. C. Mikkelsen, Jr., J. G. Santiago, Porous glass electroosmotic pumps: design and experiments. *Journal of Colloid and Interface Science* **268**, 143-153 (2003).

16. A. Alizadeh, W. L. Hsu, M. Wang, H. Daiguji, Electroosmotic flow: From microfluidics to nanofluidics. *Electrophoresis* **42**, 834-868 (2021).
17. W. Gilpin, V. N. Prakash, M. Prakash, Vortex arrays and ciliary tangles underlie the feeding–swimming trade-off in starfish larvae. *Nature Physics* **13**, 380-386 (2016).
18. D. Zhong, C. Wu, Y. Jiang, Y. Yuan, M. G. Kim, Y. Nishio, C. C. Shih, W. Wang, J. C. Lai, X. Ji, T. Z. Gao, Y. X. Wang, C. Xu, Y. Zheng, Z. Yu, H. Gong, N. Matsuhisa, C. Zhao, Y. Lei, D. Liu, S. Zhang, Y. Ochiai, S. Liu, S. Wei, J. B. Tok, Z. Bao, High-speed and large-scale intrinsically stretchable integrated circuits. *Nature* **627**, 313-320 (2024).
19. W. Wang, Y. Jiang, D. Zhong, Z. Zhang, S. Choudhury, J. C. Lai, H. Gong, S. Niu, X. Yan, Y. Zheng, C. C. Shih, R. Ning, Q. Lin, D. Li, Y. H. Kim, J. Kim, Y. X. Wang, C. Zhao, C. Xu, X. Ji, Y. Nishio, H. Lyu, J. B. Tok, Z. Bao, Neuromorphic sensorimotor loop embodied by monolithically integrated, low-voltage, soft e-skin. *Science* **380**, 735-742 (2023).
20. P. J. Flory, *Principles of Polymer Chemistry*. (Cornell University Press, 1953).
21. E. M. Purcell, "Life at Low Reynolds Number" in *Physics and Our World*. (2013), pp. 47-67.
22. E. Lauga, T. R. Powers, The hydrodynamics of swimming microorganisms. *Reports on Progress in Physics* **72**, (2009).
23. G. M. Whitesides, The origins and the future of microfluidics. *Nature* **442**, 368-373 (2006).
24. Y. W. Kim, J. Y. Yoo, Transport of solid particles in microfluidic channels. *Optics and Lasers in Engineering* **50**, 87-98 (2012).
25. M. E. Miali, W. Chien, T. L. Moore, A. Felici, A. L. Palange, M. Oneto, D. Fedosov, P. Decuzzi, Assessing Differential Particle Deformability under Microfluidic Flow Conditions. *ACS Biomater Sci Eng* **9**, 3690-3698 (2023).
26. S. Zhang, Y. Wang, P. Onck, J. den Toonder, A concise review of microfluidic particle manipulation methods. *Microfluidics and Nanofluidics* **24**, (2020).
27. S. Zhang, R. Zhang, Y. Wang, P. R. Onck, J. M. J. den Toonder, Controlled Multidirectional Particle Transportation by Magnetic Artificial Cilia. *ACS Nano* **14**, 10313-10323 (2020).
28. P. Marmottant, S. Hilgenfeldt, A bubble-driven microfluidic transport element for bioengineering. *Proc Natl Acad Sci U S A* **101**, 9523-9527 (2004).
29. M. R. Knowles, R. C. Boucher, Mucus clearance as a primary innate defense mechanism for mammalian airways. *The Journal of Clinical Investigation* **109**, 571-577 (2002).
30. R. A. Lyons, E. Saridogan, O. Djahanbakhch, The reproductive significance of human Fallopian tube cilia. *Human Reproduction Update* **12**, 363-372 (2006).
31. S. Shekhar, H. Guo, S. P. Colin, W. Marshall, E. Kanso, J. H. Costello, Cooperative hydrodynamics accompany multicellular-like colonial organization in the unicellular ciliate *Stentor*. *Nature Physics* **21**, 624-631 (2025).
32. D. B. Hill, V. Swaminathan, A. Estes, J. Cribb, E. T. O'Brien, C. W. Davis, R. Superfine, Force generation and dynamics of individual cilia under external loading. *Biophys J* **98**, 57-66 (2010).
33. K. A. Schmitz, D. L. Holcomb-Wygle, D. J. Oberski, C. B. Lindemann, Measurement of the force produced by an intact bull sperm flagellum in isometric arrest and estimation of the dynein stall force. *Biophys J* **79**, 468-478 (2000).

34. C. Y. Li, S. Y. Zheng, X. P. Hao, W. Hong, Q. Zheng, Z. L. Wu, Spontaneous and rapid electro-actuated snapping of constrained polyelectrolyte hydrogels. *Science Advances* **8**, eabm9608 (2022).
35. M. Chatenet, B. G. Pollet, D. R. Dekel, F. Dionigi, J. Deseure, P. Millet, R. D. Braatz, M. Z. Bazant, M. Eikerling, I. Staffell, P. Balcombe, Y. Shao-Horn, H. Schafer, Water electrolysis: from textbook knowledge to the latest scientific strategies and industrial developments. *Chem Soc Rev* **51**, 4583-4762 (2022).
36. E. A. Kamoun, A. El-Betany, H. Menzel, X. Chen, Influence of photoinitiator concentration and irradiation time on the crosslinking performance of visible-light activated pullulan-HEMA hydrogels. *Int J Biol Macromol* **120**, 1884-1892 (2018).
37. M. Kedzierska, M. Bankosz, P. Potemski, Studies on the Impact of the Photoinitiator Amount Used during the PVP-Based Hydrogels' Synthesis on Their Physicochemical Properties. *Materials (Basel)* **15**, (2022).